# A Graph is Worth 1-bit Spikes: When Graph Contrastive Learning Meets Spiking Neural Networks

**Jintang Li**[1]*, **Huizhe Zhang**[1], **Ruofan Wu**[2], **Zulun Zhu**[3], **Baokun Wang**[2],
**Changhua Meng**[2], **Zibin Zheng**[1], **Liang Chen**[1]†
[1]Sun Yat-sen University
[2]Ant Group
[3]Nanyang Technological University
`{lijt55,zhanghzh33}@mail2.sysu.edu.cn;`
`{ruofan.wrf,yike.wbk,changhua.mch}@antgroup.com;`
`ZULUN001@e.ntu.edu.sg`

## Abstract

While contrastive self-supervised learning has become the de-facto learning paradigm for graph neural networks, the pursuit of higher task accuracy requires a larger hidden dimensionality to learn informative and discriminative full-precision representations, raising concerns about computation, memory footprint, and energy consumption burden (largely overlooked) for real-world applications. This work explores a promising direction for graph contrastive learning (GCL) with spiking neural networks (SNNs), which leverage sparse and binary characteristics to learn more biologically plausible and compact representations. We propose SPIKEGCL, a novel GCL framework to learn binarized 1-bit representations for graphs, making balanced trade-offs between efficiency and performance. We provide theoretical guarantees to demonstrate that SPIKEGCL has comparable expressiveness with its full-precision counterparts. Experimental results demonstrate that, with nearly 32x representation storage compression, SPIKEGCL is either comparable to or outperforms many fancy state-of-the-art supervised and self-supervised methods across several graph benchmarks.

## 1 Introduction

Graph neural networks (GNNs) have demonstrated remarkable capabilities in learning representations of graphs and manifolds that are beneficial for a wide range of tasks. Especially since the advent of recent self-supervised learning techniques, rapid progress toward learning universally useful representations has been made Liu et al. (2021); Yu et al. (2021a;b). Graph contrastive learning (GCL), which aims to learn generalizable and transferable graph representations by contrasting positive and negative sample pairs taken from different graph views, has become the hotspot in graph self-supervised learning. As an active area of research, numerous variants of GCL methods have been proposed to achieve state-of-the-art performance in graph-based learning tasks, solving the dilemma of learning useful representations from graph data without end-to-end supervision Li et al. (2023a); Velickovic et al. (2019); Thakoor et al. (2021); Zheng et al. (2022).

In recent years, real-world graphs have been scaling and growing even larger. For example, Amazon's product recommendation graph has over 150M users and 350M products Chen et al. (2022b), while Microsoft Academic Graph consists of more than 120M publications and related authors, venues, organizations, and fields of study Sinha et al. (2015). New challenges beyond label annotations have arisen in terms of processing, storage, and deployment in many industrial scenarios Chen et al. (2022a). While GNNs-based contrastive learning has advanced the frontiers in many applications, current state-of-the-arts mainly requires large hidden dimensions to learn generalizable *full-precision*

---

*Work done during an internship at Ant Group.
†Corresponding author.

representations Mo et al. (2022); Li et al. (2023a), making them memory inefficient, storage excessive, and computationally expensive especially for resource-constrained edge devices Zhou et al. (2023).

In parallel, biological neural networks continue to inspire breakthroughs in modern neural network performance, with prominent examples including spiking neural networks (SNNs) Malcolm & Casco-Rodriguez (2023). SNNs are a class of brain-inspired networks with asynchronous discrete and sparse characteristics, which have increasingly manifested their superiority in low energy consumption and inference latency Feng et al. (2022). Unlike traditional artificial neural networks (ANNs), which use floating-point outputs, SNNs use a more biologically plausible approach where neurons communicate via sparse and binarized representations, referred to as 'spikes'. Such characteristics make them a promising choice for low-power, mobile, or otherwise hardware-constrained settings. In literature Zhou et al. (2022); Zhu et al. (2022), SNNs are proven to consume ∼100x less energy compared with modern ANNs on the neuromorphic chip (e.g. ROLLs Indiveri et al. (2015)).

Inspired by the success of vision research Kim et al. (2021); Zhou et al. (2022), some recent efforts have been made toward generalizing SNNs to graph data. For example, SpikingGCN Zhu et al. (2022) and SpikeNet Li et al. (2023b) are two representative works combining SNNs with GNN architectures, in which SNNs are utilized as the basic building blocks to model the underlying graph structure and dynamics. Despite the promising potentiality, SNNs have been under-appreciated and under-investigated in graph contrastive learning. This has led to an interesting yet rarely explored research question:

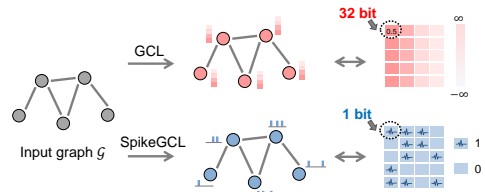

*Can we explore the possibilities of SNNs with contrastive learning schemes to learn sparse, binarized yet generalizable representations?*

Figure 1: Comparison between conventional GCL and SPIKEGCL. Instead of using full-precision (32-bit) representations, SPIKEGCL produces sparse and compact 1-bit representations, making them more memory-friendly and computationally efficient for cheap devices with limited resources.

**Present work.** Deviating from the large body of prior works on graph contrastive learning, in this paper we take a new perspective to address self-supervised and binarized representation learning on graphs. We present SPIKEGCL, a principled GCL framework built upon SNNs to learn binarized graph representations in a compact and efficient manner. Instead of learning full-precision node representations, we learn sparse and compact 1-bit representations to unlock the power of SNNs on graph data and enable fast inference. In addition, we noticed the problem of vanishing gradients (surprisingly overlooked) during direct training of SNNs and proposed a blockwise training approach to tackle it. Although SPIKEGCL uses binarized 1-bit spikes as representations, it comes with both theoretical guarantees and comparable performance in terms of expressiveness and efficacy, respectively. Compared with floating point counterparts, SPIKEGCL leads to less storage space, smaller memory footprint, lower power consumption, and faster inference speed, all of which are essential for practical applications such as edge deployment.

Overall, our contributions can be mainly summarized as follows:

- We believe our work is timely. In response to the explosive data expansion, we study the problem of self-supervised and binarized representations learning on graphs with SNNs, whose potential is especially attractive for low-power and resource-constrained settings.

- We propose SPIKEGCL to learn binarized representations from large-scale graphs. SPIKEGCL exhibits high hardware efficiency and significantly reduces memory consumption of node representations by ∼32x and energy consumption of GNNs by up to ∼7x, respectively. SPIKEGCL is a theoretically guaranteed framework with powerful capabilities in learning node representations.

- We address the challenge of training deep SNNs with a simple blockwise learning paradigm. By limiting the backpropagation path to a single block of time steps, we prevent SNNs from suffering from the notorious problem of vanishing gradients.

- Extensive experiments demonstrate that SPIKEGCL performs on par with or even sometimes better than advanced full-precision competitors, across multiple graphs of various scales. Our results hold great promise for accurate and fast online inference of graph-based systems.

We leave two additional remarks: (i) SPIKEGCL is designed to reduce computation, energy, and storage costs from the perspective of compressing graph representations using biological networks, which is orthogonal to existing advances on sampling Hamilton et al. (2017), data condensation Jin et al. (2022); Zeng et al. (2020) and architectural simplifications Thakoor et al. (2021); Mo et al. (2022); Zheng et al. (2022). (ii) To the best of our knowledge, our work is the first to explore the feasibility of implementing graph self-supervised learning with SNNs for learning binarized representations. We hope our work will inspire the community to explore more promising algorithms.

## 2 RELATED WORK

**Graph contrastive learning.** Self-supervised representation learning from graph data is a quickly evolving field. The last years have witnessed the emergence of a promising self-supervised learning strategy, referred to as graph contrastive learning (GCL) Liu et al. (2021). Typically, GCL works by contrasting so-called positive samples against negative ones from different graph augmentation views, which has led to the development of several novel techniques and frameworks such as DGI Velickovic et al. (2019), GraphCL You et al. (2020), GRACE Zhu et al. (2020), and GGD Zheng et al. (2022). Recent research has also suggested that negative samples are not always necessary for graph contrastive learning, with prominent examples including BGRL Thakoor et al. (2021) and CCA-SSG Zhang et al. (2021). Negative-sample-free approach paves the way to a simple yet effective GCL method and frees the model from intricate designs. We refer readers to Liu et al. (2021) for a comprehensive review of graph contrastive learning. Despite the achievements, current GCL designs mainly focus on task accuracy with a large hidden dimensionality Mo et al. (2022); Li et al. (2023a), and lack consideration of hardware resource limitations to meet the real-time requirements of edge application scenarios Zhou et al. (2023).

**Spiking neural networks.** SNNs, which mimic biological neural networks by leveraging sparse and event-driven activations, are very promising for low-power applications. SNNs have been around for a while, and although they have not gained the same level of popularity as GNNs, they have steadily increased their influence in recent years. Over the past few years, SNNs have gained significant popularity in the field of vision tasks, including image classification Zhou et al. (2022), objection detection Kim et al. (2020), and segmentation Kim et al. (2021). Efforts have been made to incorporate their biological plausibility and leverage their promising energy efficiency into GNN learning. SpikingGCN Zhu et al. (2022) is a recent endeavor on encoding the node representation through SNNs. As SNNs are intrinsically dynamic with abundant temporal information conveyed by spike timing, SpikeNet Li et al. (2023c) then proceeds to model the temporal evolution of dynamic graphs via SNNs. In spite of the increasing research interests, the benefits of SNNs have not been discovered in GCL yet.

**Binarized graph representation learning.** Model size, memory footprint, and energy consumption are common concerns for many real-world applications Bahri et al. (2021). In reaction, binarized graph representation learning has been developed in response to the need for more efficient and compact representations for learning on large-scale graph data. Currently, there are two lines of research being pursued in this area. One line of research involves directly quantizing graphs by using discrete hashing techniques Yang et al. (2018); Bahri et al. (2021) or estimating gradients of non-differentiable quantization process Chen et al. (2022b). Another research line to produce binary representations using deep learning methods are those binarized networks, which are designed for fast inference and small memory footprint, in which binary representation is only a by-product Wang et al. (2021). On one hand, binarized graph representation learning is a novel and promising direction that aims to encode graphs into compact binary vectors that can facilitate efficient storage, retrieval, and computation. On the other hand, SNNs that use discrete spikes rather than real values to communicate between neurons are naturally preferred for learning binary representations. Therefore, it is intuitively promising to explore the marriage between them.

## 3 PRELIMINARIES

**Problem formulation.** Let $\mathcal{G} = (\mathcal{V}, \mathcal{E}, \mathbf{X})$ denote an attributed undirected graph where $\mathcal{V} = \{v_i\}_{i=1}^N$ and $\mathcal{E} \subseteq \mathcal{V} \times \mathcal{V}$ are a set of nodes and edges, respectively. We focus primarily on undirected graphs though it is straightforward to extend our study to directed graphs. $\mathcal{G}$ is associated with an

$d$-dimensional attribute feature matrix $\mathbf{X} = \{x_i\}_{i=1}^N \in \mathbb{R}^{N \times d}$. In the self-supervised learning setting of this work, our objective is to learn an encoder $f_\theta : \mathbb{R}^d \to \{0,1\}^d$ parameterized by $\theta$, which maps between the space of graph $\mathcal{G}$ and their low-dimensional and *binary* latent representations $\mathbf{Z} = \{z_i\}_{i=1}^N$, such that $f_\theta(\mathcal{G}) = \mathbf{Z} \in \{0,1\}^{N \times d}$ given $d$ the embedding dimension. Note that for simplicity we assume the feature dimensions are the same across all layers.

**Spiking neural networks.** Throughout existing SNNs, the integrate-fire (IF) Salinas & Sejnowski (2002) model and its variants are commonly adopted to formulate the spiking neuron and evolve into numerous variants with different biological features. As the name suggests, IF models have three fundamental characteristics: (i) **Integrate**. The neuron integrates current by means of the capacitor over time, which leads to a charge accumulation; (ii) **Fire**. When the membrane potential has reached or exceeded a given threshold $V_{\text{th}}$, it fires (i.e., emits a spike). (iii) **Reset**. After that, the membrane potential is reset, and here we introduce two types of reset Rueckauer et al. (2016): *reset to zero* which always sets the membrane potential back to a constant value $V_{\text{reset}} < V_{\text{th}}$, typically zero, whereas *reset by subtraction* subtracts the threshold $V_{\text{th}}$ from the membrane potential at the time where the threshold is exceeded: We use a unified model to describe the dynamics of IF-based spiking neurons:

$$\textbf{Integrate:} \qquad V^t = \Psi(V^{t-1}, I^t), \tag{1}$$

$$\textbf{Fire:} \qquad S^t = \Theta(V^t - V_{\text{th}}), \tag{2}$$

$$\textbf{Reset:} \qquad V^t = \begin{cases} S^t V_{\text{reset}} + (1 - S^t)V^t, & \text{reset to zero}, \\ S^t(V^t - V_{\text{th}}) + (1 - S^t)V^t, & \text{reset by subtraction}, \end{cases} \tag{3}$$

where $I^t$ and $V^t$ denote the input current and membrane potential (voltage) at time-step $t$, respectively. The decision to fire a spike in the neuron output is carried out according to the Heaviside step function $\Theta(\cdot)$, which is defined by $\Theta(x) = 1$ if $x \geq 0$ and 0 otherwise. The function $\Psi(\cdot)$ in Eq. 1 describes how the spiking neuron receives the resultant current and accumulates membrane potential. We have IF Salinas & Sejnowski (2002) and its variant Leaky Integrate-and-Fire (LIF) Gerstner et al. (2014) model, formulated as follows:

$$\textbf{IF:} \qquad V^t = V^{t-1} + I^t, \tag{4}$$

$$\textbf{LIF:} \qquad V^t = V^{t-1} + \frac{1}{\tau_m}\left(I^t - (V^{t-1} - V_{\text{reset}})\right), \tag{5}$$

where $\tau_m$ in Eq. 5 represents the membrane time constant to control how fast the membrane potential decays, leading to the membrane potential charges and discharges exponentially in response to the current inputs. Typically, $\tau_m$ can also be optimized automatically instead of manually tuning to learn different neuron dynamics during training, which we referred to as Parametric LIF (PLIF) Fang et al. (2020). In this paper, the surrogate gradient method is used to define $\Theta'(x) \triangleq \sigma'(\alpha x)$ during error back-propagation, with $\sigma(\cdot)$ denote the surrogate function such as Sigmoid and $\alpha$ the smooth factor Li et al. (2023b).

# 4 SPIKING GRAPH CONTRASTIVE LEARNING (SPIKEGCL)

In this section, we introduce our proposed SPIKEGCL framework for learning binarized representations. SPIKEGCL is a simple yet effective derivative of the standard GCL framework with minimal but effective modifications on the encoder design coupled with SNNs. In what follows, we first shed light on building sequential inputs for SNNs from a single non-temporal graph (§ 4.1) and depict how to binarize node representations using SNNs (§ 4.2). Then, we explain the architectural overview and detailed components of SPIKEGCL one by one (§ 4.3). Finally, we employ blockwise learning for better training of deep SNNs (§ 4.4).

## 4.1 GROUPING NODE FEATURES

Typically, SNNs require sequential inputs to perform the integrated-and-fire process to emit spikes. One major challenge is how to formulate such inputs from a non-temporal graph. A common practice introduced in literature is to repeat the graph multiple times (i.e., a given time window $T$), typically followed by probabilistic encoding methods (e.g., Bernoulli encoding) to generate diverse input graphs Zhu et al. (2022); Xu et al. (2021). However, this will inevitably introduce high computational and memory overheads, becoming a major bottleneck for SNNs to scale to large graphs.

In this work, we adopt the approach of partitioning the graph data into different groups rather than repeating the graph multiple times to construct the sequential inputs required for SNNs. Given a time window $T$ $(T > 1)$ and a graph $\mathcal{G}$, we uniformly partition the node features into the following $T$ groups[1]: $\mathbf{X} = [\mathbf{X}^1, \ldots, \mathbf{X}^T]$ along the feature dimension, where $\mathbf{X}^t \in \mathbb{R}^{N \times \frac{d}{T}}$ consists of the group of features in the $t$-th partition. In cases where $d$ cannot be divided by $T$, we have $\mathbf{X}^t \in \mathbb{R}^{N \times (d//T)}$ for $t < T$ and $\mathbf{X}^T \in \mathbb{R}^{N \times (d \bmod T)}$. Thus we have $T$ subgraphs as

$$\hat{\mathcal{G}} = [\mathcal{G}^1, \ldots, \mathcal{G}^T] = [(\mathcal{V}, \mathcal{E}, \mathbf{X}^1), \ldots, (\mathcal{V}, \mathcal{E}, \mathbf{X}^T)]. \tag{6}$$

Each subgraph in $\hat{\mathcal{G}}$ shares the same graph structure but only differs in node features. Note that $[\mathbf{X}^1, \ldots, \mathbf{X}^T]$ are *non-overlapping* groups, which avoids unnecessary computation on input features. Our strategy leads to huge memory and computational benefits as each subgraph in the graph sequence $\hat{\mathcal{G}}$ only stores a subset of the node features instead of the whole set. This is a significant improvement over previous approaches such as SpikingGCN Zhu et al. (2022), as it offers substantial benefits in terms of computational and memory complexity.

## 4.2 BINARIZING GRAPH REPRESENTATIONS

In GCL, GNNs are widely adopted as encoders for representing graph-structured data. GNNs generally follow the canonical *message passing* scheme in which each node's representation is computed recursively by aggregating representations ('messages') from its immediate neighbors Kipf & Welling (2016); Hamilton et al. (2017). Given an $L$ layer GNN $f_\theta$, the updating process of the $l$-th layer could be formulated as:

$$h_u^{(l)} = \text{COMBINE}^{(l)}\left(\left\{h_u^{(l-1)}, \text{AGGREGATE}^{(l)}\left(\left\{h_v^{(l-1)} : v \in \mathcal{N}_u\right\}\right)\right\}\right) \tag{7}$$

where $\text{AGGREGATE}(\cdot)$ is an aggregation function that aggregates features of neighbors, and $\text{COMBINE}(\cdot)$ denotes the combination of aggregated features from a central node and its neighbors. $h_u^{(l)}$ is the embedding of node $u$ at the $l$-th layer of GNN, where $l \in \{1, \ldots, L\}$ and initially $h_u^{(0)} = x_u$; $\mathcal{N}_u$ is the set of neighborhoods of $u$. After $L$ rounds of aggregation, each node $u \in \mathcal{V}$ obtains its representation vector $h_u^{(L)}$. The final node representation is denoted as $\mathbf{H} = [h_1^{(L)}, \ldots, h_N^{(L)}]$.

We additionally introduce SNNs to the encoder for binarizing node representations. SNNs receive continuous values and convert them into binary spike trains, which opens up possibilities for mapping graph structure from continuous Euclidian space into discrete and binary one. Given a graph sequence $\hat{\mathcal{G}} = [\mathcal{G}^1, \ldots, \mathcal{G}^T]$, we form $T$ graph encoders $[f_{\theta_1}^1, \ldots, f_{\theta_T}^T]$ accordingly such that each encoder $f_{\theta_t}^t$ maps a graph $\mathcal{G}_t$ to its hidden representations $\mathbf{H}^t$. Here we denote $f_\theta = [f_{\theta_1}^1, \ldots, f_{\theta_T}^T]$ with slightly abuse of notations. Then, a spiking neuron is employed on the outputs to generate binary spikes in a dynamic manner:

$$\mathbf{S}^t = \Theta\left(\Psi(V^{t-1}, \mathbf{H}^t) - V_{\text{th}}\right), \quad \mathbf{H}^t = f_\theta^t(\mathcal{G}^t), \tag{8}$$

where $\Psi(\cdot)$ denotes a spiking function that receives the resultant current from the encoder and accumulates membrane potential to generate the spikes, such as IF or LIF introduced in § 3. $\mathbf{S}^t \in \{0, 1\}^{N \times \frac{d}{T}}$ is the output spikes at time step $t$. In this regard, the binary representations are derived by taking historical spikes in each time step with a *concatenate* pooling Li et al. (2023b), i.e., $\mathbf{Z} = \left(\mathbf{S}^1 || \cdots || \mathbf{S}^T\right)$.

## 4.3 OVERALL FRAMEWORK

We decompose our proposed SPIKEGCL from four dimensions: (i) augmentation, (ii) encoder, (iii) predictor head (a.k.a. decoder), and (iv) contrastive objective. These four components constitute the design space of interest in this work. We present an architectural overview of SPIKEGCL in Figure 2(a), as well as the core step of embedding generation in Figure 2(b).

**Augmentation.** Augmentation is crucial for contrastive learning by providing different graph views for contrasting. With a given graph $\mathcal{G}$, SPIKEGCL involves bi-level augmentation techniques, i.e.,

---

[1]Exploring non-uniform partition with clustering methods is an important direction for future work.

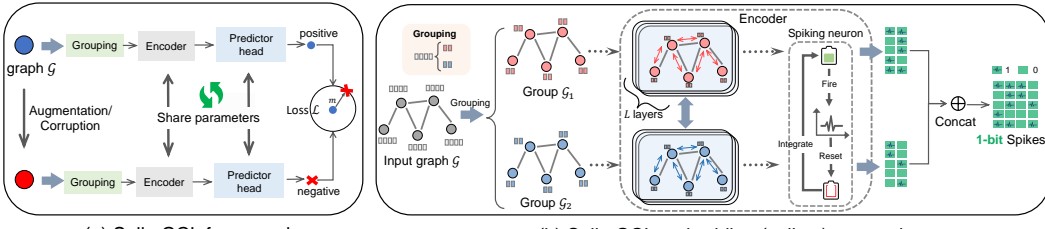

(a) SpikeGCL framework          (b) SpikeGCL embedding (spikes) generation

Figure 2: Overview of SPIKEGCL framework. **(a)** SPIKEGCL follows the standard GCL philosophy, which learns binary representations by contrasting positive and negative samples with a margin ranking loss. **(b)** SPIKEGCL first partitions node features into $T$ non-overlapping groups, each of which is then fed to an encoder whereby spiking neurons represent nodes of graph as 1-bit spikes.

topology (structure) and feature augmentation, to construct its corrupted version $\tilde{\mathcal{G}}$. To be specific, we harness two ways for augmentation: *edge dropping* and *feature shuffling*. Edge dropping randomly drops a subset of edges from the original graph, while feature shuffling gives a new permutation of the node feature matrix along the feature dimension. In this way, we obtain the corrupted version of an original graph $\tilde{\mathcal{G}} = \{\mathcal{V}, \tilde{\mathcal{E}}, \tilde{\mathbf{X}}\}$, where $\tilde{\mathcal{E}} \subseteq \mathcal{E}$ and $\tilde{\mathbf{X}}$ denotes the column-wise shuffled feature matrix such that $\tilde{\mathbf{X}} = \mathbf{X}[:, \mathcal{P}]$ with $\mathcal{P}$ the new permutation of features. Since we partition node features in a sequential manner that is sensitive to the permutation of input features, feature shuffling is able to offer *hard negative* samples for contrastive learning.

**Encoder.** As detailed in § 4.2, our encoder is a series of peer GNNs corresponding to each group of input node features, followed by a spiking neuron to binarize the representations. Among the many variants of GNNs, GCN Kipf & Welling (2016) is the most representative structure, and we adopt it as the basic unit of the encoder in this work. The number of peer GNNs in the encoder is relative to the number of time steps $T$, which makes the model excessively complex and potentially lead to overfitting if $T$ is large. We circumvent this problem by **parameter sharing**. Note that only the first layer is different in response to diverse groups of features. For the remaining $L-1$ layers, parameters are shared across peer GNNs to prevent excessive memory footprint and the overfitting issue.

**Predictor head (decoder).** A non-linear transformation named projection head is adopted to map binarized representations to continuous latent space where the contrastive loss is calculated, as advocated in Thakoor et al. (2021); Zhang et al. (2021). As a proxy for learning on discrete spikes, we employ a single-layer perceptron (MLP) to the learned representations, i.e., $g_\phi(z_u) = \text{MLP}(z_u), \forall u \in \mathcal{V}$. Since the hidden representations are binary, we can instead use 'masked summation' Li et al. (2023b) to enjoy the sparse characteristics and avoid expensive matrix multiplication computations during training and inference.

**Contrastive objective.** Contrastive objectives are widely used to measure the similarity or distance between positive and negative samples. Rather than explicitly maximizing the discrepancy between positive and negative pairs as most existing works on contrastive learning have done, the 'contrastiveness' in this approach is reflected in the diverse 'distance' naturally measured by a parameterized model $g_\phi(\cdot)$. Specifically, we employ a margin ranking loss (MRL) Chen et al. (2020) to $g_\phi(\cdot)$:

$$\mathcal{J} = \frac{1}{N} \sum_{u \in \mathcal{V}} \max(0, g_\phi(z_u) - g_\phi(z_u^-) + m), \tag{9}$$

with the margin $m$ a hyperparameter to make sure the model disregards abundant far (easy) negatives and leverages scarce nearby (hard) negatives during training. $z_u$ and $z_u^-$ is the representation of node $u$ obtained from original graph sequence $\hat{\mathcal{G}}$ and its corrupted one, respectively. MRL forces the score of positive samples to be lower (towards zero) and assigns a higher score to negative samples by a margin of at least $m$. Therefore, positive samples are separated from negative ones.

### 4.4 BLOCKWISE SURROGATE GRADIENT LEARNING

From an optimization point of view, SNNs lack straightforward gradient calculation for backward propagation, and also methods that effectively leverage their inherent advantages. Recent attempts

have been made to surrogate gradient learning Lee et al. (2016), an alternative of gradient descent that avoids the non-differentiability of spike signals. As a backpropagation-like training method, it approximates the backward gradients of the hard threshold function using a smooth activation function (such as Sigmoid) during backpropagation. At present, the surrogate learning technique plays an extremely important role in advanced methods to learn SNNs properly Zhu et al. (2022); Xu et al. (2021); Li et al. (2023b); Fang et al. (2021; 2020).

Despite the promising results, surrogate gradient learning has its own drawbacks. Typically, SNNs require relatively large time steps to approximate continuous inputs and endow the network with better expressiveness Zhu et al. (2022). However, a large time step often leads to many problems such as high overheads and network degradation Fang et al. (2021). Particularly, the training of SNNs also comes with a serious vanishing gradient problem in which gradients quickly diminish with time steps. We have further discussion with respect to the phenomenon in Appendix B. These drawbacks greatly limit the performance of directly trained SNNs and prevent them from going 'deeper' with long sequence inputs.

In this paper, we explore alternatives to end-to-end back-propagation in the form of surrogate gradient learning rules, leveraging the latest advances in self-supervised learning Siddiqui et al. (2023) to address the above limitation in SNNs. Specifically, we propose a blockwise training strategy that separately learns each block with a local objective. We consider one or more consecutive time steps as a single block, and limit the length of the backpropagation path to each of these blocks. Parameters within each block are optimized locally with a contrastive learning objective, using stop-gradient to prevent the gradients from flowing through the blocks. A technical comparison between the existing end-to-end surrogate learning paradigm and our proposed local training strategy is shown in Figure 3.

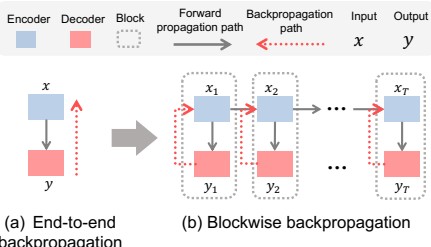

(a) End-to-end backpropagation  (b) Blockwise backpropagation

Figure 3: A comparison between two back-propagation learning paradigms. The back-propagation path during blockwise training is limited to a single block of networks to avoid a large memory footprint and vanishing gradient problem.

## 5 THEORETICAL GUARANTEES

There are several works on designing practical graph-based SNNs that achieve comparable performance with GNNs Li et al. (2023c); Xu et al. (2021); Zhu et al. (2022). However, the basic principles and theoretical groundwork for their performance guarantees are lacking, and related research only shows the similarity between the IF neuron model and the ReLU activation Rueckauer et al. (2016). In this section, we are motivated to bridge the gap between SNNs and GNNs. We present an overview of theoretical results regarding the approximation properties of SPIKEGCL with respect to its full-precision counterparts through the following theorem:

**Theorem 1** (Informal). *For any full-precision GNN with a hidden dimension of $d/T$, there exists a corresponding SPIKEGCL such that its approximation error, defined as the $\ell_2$ distance between the firing rates of the SPIKEGCL representation and the GNN representation at any single node, is of the order $\Theta(1/T)$.*

In the order relation $\Theta(\frac{1}{T})$, we hide factors dependent upon the underlying GNN structure, network depth as well as characteristics of the input graph, which will be stated precisely in Theorem Theorem 2 in Appendix A. Theorem 1 suggests that with a sufficiently large node feature dimension such that we may set a large simulation length $T$, we are able to (almost) implement vanilla GNNs with a much better computational and energy efficiency. Moreover, the approximation is defined using the firing rates of SNN outputs, which measures only a restricted set of inductive biases offered by SPIKEGCL. Consequently, we empirically observe that a moderate level of $T$ might also provide satisfactory performance in our experiments. Our analysis is presented to shed insights on the connection between the computational neuroscience model (e.g., SNNs) and the machine learning neural network model (e.g., GNNs). This connection has been analytically proven under certain conditions and empirically demonstrated through our experiments (see § 6). It can serve as the theoretical basis for potentially combining the respective merits of the two types of neural networks.

## 6 EXPERIMENTS

In this section, we perform experimental evaluations to demonstrate the effectiveness of our proposed SPIKEGCL framework. Due to space limitations, we present detailed experimental settings and additional results in Appendix E and Appendix F.

**Datasets.** We evaluate SPIKEGCL on several graph benchmarks with different scales and properties. Following prior works Zhang et al. (2021); Thakoor et al. (2021); Zhu et al. (2022), we adopt 9 common benchmark graphs, including two co-purchase graphs, i.e., Amazon-Photo, Amazon-Computer Shchur et al. (2018), two co-author graphs, i.e., Coauthor-CS and Coauthor-Physics Shchur et al. (2018), three citation graphs, i.e., Cora, CiteSeer, PubMed Sen et al. (2008), as well as two large-scale datasets ogbn-arXiv and ogbn-MAG from Open Graph Benchmark Hu et al. (2020). The detailed introduction and statistics of these datasets are presented in Appendix E.

**Baselines.** We compare our proposed methods to a wide range of baselines that fall into four categories: (i) full-precision (supervised) GNNs: GCN Kipf & Welling (2016) and GAT Veličković et al. (2018); (ii) 1-bit quantization-based GNNs: Bi-GCN Wang et al. (2021), BinaryGNN Bahri et al. (2021) and BANE Yang et al. (2018); (iii) contrastive methods: DGI Velickovic et al. (2019), GRACE Zhu et al. (2020), CCA-SSG Zhang et al. (2021), BGRL Thakoor et al. (2021), SUGRL Mo et al. (2022), and GGD Zheng et al. (2022); (iv) Graph SNNs: SpikingGCN Zhu et al. (2022), SpikeNet Li et al. (2023b), GC-SNN and GA-SNN Xu et al. (2021). SpikeNet is initially designed for dynamic graphs, we adapt the author's implementation to static graphs following the practice in Zhu et al. (2022); Xu et al. (2021). The hyperparameters of all the baselines were configured according to the experimental settings officially reported by the authors and were then carefully tuned in our experiments to achieve their best results.

Table 1: Classification accuracy (%) on six large scale datasets. The best result for each dataset is highlighted in **red**. The missing results are due to the out-of-memory error on a GPU with 24GB memory. (**U**: unsupervised or self-supervised; **S**: spike-based; **B**: binarized)

| | U | S | B | Computers | Photo | CS | Physics | arXiv | MAG |
|---|---|---|---|---|---|---|---|---|---|
| GCN Kipf & Welling (2016) | | | | $86.5_{\pm 0.5}$ | $92.4_{\pm 0.2}$ | $92.5_{\pm 0.4}$ | $95.7_{\pm 0.5}$ | $70.4_{\pm 0.3}$ | $30.1_{\pm 0.3}$ |
| GAT Veličković et al. (2018) | | | | $86.9_{\pm 0.2}$ | $92.5_{\pm 0.3}$ | $92.3_{\pm 0.2}$ | $95.4_{\pm 0.3}$ | $70.6_{\pm 0.3}$ | $30.5_{\pm 0.3}$ |
| SpikeNet Li et al. (2023b) | | ✓ | | $88.0_{\pm 0.7}$ | $92.9_{\pm 0.1}$ | $\textcolor{red}{93.4}_{\pm 0.2}$ | $\textcolor{red}{95.8}_{\pm 0.7}$ | $66.8_{\pm 0.1}$ | - |
| SpikingGCN Zhu et al. (2022) | | ✓ | | $86.9_{\pm 0.3}$ | $92.6_{\pm 0.7}$ | $92.6_{\pm 0.3}$ | $94.3_{\pm 0.1}$ | $55.8_{\pm 0.7}$ | - |
| GC-SNN Xu et al. (2021) | | ✓ | | $88.2_{\pm 0.6}$ | $92.8_{\pm 0.1}$ | $93.0_{\pm 0.4}$ | $95.6_{\pm 0.7}$ | - | - |
| GA-SNN Xu et al. (2021) | | ✓ | | $88.1_{\pm 0.1}$ | $\textcolor{red}{93.5}_{\pm 0.6}$ | $92.2_{\pm 0.1}$ | $95.8_{\pm 0.5}$ | - | - |
| Bi-GCN Wang et al. (2021) | | | ✓ | $86.4_{\pm 0.3}$ | $92.1_{\pm 0.9}$ | $91.0_{\pm 0.7}$ | $93.3_{\pm 1.1}$ | $66.0_{\pm 0.8}$ | $28.2_{\pm 0.4}$ |
| BinaryGNN Bahri et al. (2021) | | | ✓ | $87.8_{\pm 0.2}$ | $92.4_{\pm 0.2}$ | $91.2_{\pm 0.1}$ | $95.3_{\pm 0.1}$ | $67.2_{\pm 0.9}$ | - |
| BANE Yang et al. (2018) | ✓ | | ✓ | $72.7_{\pm 0.3}$ | $78.2_{\pm 0.3}$ | $92.8_{\pm 0.1}$ | $93.4_{\pm 0.4}$ | >3days | >3days |
| DGI Velickovic et al. (2019) | ✓ | | | $84.0_{\pm 0.5}$ | $91.6_{\pm 0.2}$ | $92.2_{\pm 0.6}$ | $94.5_{\pm 0.5}$ | $65.1_{\pm 0.4}$ | $31.4_{\pm 0.3}$ |
| GRACE Zhu et al. (2020) | ✓ | | | $86.3_{\pm 0.3}$ | $92.2_{\pm 0.2}$ | $92.9_{\pm 0.0}$ | $95.3_{\pm 0.0}$ | $68.7_{\pm 0.4}$ | $31.5_{\pm 0.3}$ |
| CCA-SSG Zhang et al. (2021) | ✓ | | | $88.7_{\pm 0.3}$ | $93.1_{\pm 0.1}$ | $93.3_{\pm 0.1}$ | $95.7_{\pm 0.1}$ | $71.2_{\pm 0.2}$ | $31.8_{\pm 0.4}$ |
| BGRL Thakoor et al. (2021) | ✓ | | | $\textcolor{red}{90.3}_{\pm 0.2}$ | $93.2_{\pm 0.3}$ | $93.3_{\pm 0.1}$ | $95.7_{\pm 0.0}$ | $71.6_{\pm 0.1}$ | $31.1_{\pm 0.1}$ |
| SUGRL Mo et al. (2022) | ✓ | | | $88.9_{\pm 0.2}$ | $93.2_{\pm 0.4}$ | $93.4_{\pm 0.0}$ | $95.2_{\pm 0.0}$ | $68.8_{\pm 0.4}$ | $\textcolor{red}{32.4}_{\pm 0.1}$ |
| GGD Zheng et al. (2022) | ✓ | | | $88.0_{\pm 0.1}$ | $92.9_{\pm 0.2}$ | $93.1_{\pm 0.1}$ | $95.3_{\pm 0.0}$ | $\textcolor{red}{71.6}_{\pm 0.5}$ | $31.7_{\pm 0.7}$ |
| SPIKEGCL | ✓ | ✓ | ✓ | $88.9_{\pm 0.3}$ | $93.0_{\pm 0.1}$ | $92.8_{\pm 0.1}$ | $95.2_{\pm 0.6}$ | $70.9_{\pm 0.1}$ | $32.0_{\pm 0.3}$ |

**Overall performance.** The results on six graph datasets are summarized in Table 1. We defer the results on Cora, CiteSeer, and PubMed to Appendix F. Table 1 shows a significant performance gap between full-precision methods and 1-bit GNNs, particularly with the unsupervised method BANE. Even supervised methods Bi-GCN and binaryGNN struggle to match the performance of full-precision methods. In contrast, SPIKEGCL competes comfortably and sometimes outperforms advanced full-precision methods. When compared to full-precision GCL methods, our model approximates about 95%~99% of their performance capability across all datasets. Additionally, SPIKEGCL performs comparably to supervised graph SNNs and offers the possibility to scale to large graph datasets (e.g., arXiv and MAG). Overall, the results indicate that binary representation does not necessarily lead to accuracy loss as long as it is properly trained.

**Efficiency.** We first provide a comparison of the number of parameters and theoretical energy consumption of SPIKEGCL and full-precision GCL methods. The computation of theoretical energy

Table 2: The parameter size (KB) and theoretical energy consumption (mJ) of various GCL methods. The row in 'Average' denotes the averaged results of full-precision GCL baselines. Darker color in SPIKEGCL indicates a larger improvement in efficiency over the baselines.

| | Computers | | CS | | Physics | | arXiv | | MAG | |
|---|---|---|---|---|---|---|---|---|---|---|
| | #Param↓ | Energy↓ | #Param↓ | Energy↓ | #Param↓ | Energy↓ | #Param↓ | Energy↓ | #Param↓ | Energy↓ |
| DGI | 917.5 | 0.5 | 4008.9 | 8 | 4833.3 | 6 | 590.3 | 5 | 590.3 | 568 |
| GRACE | 656.1 | 1.1 | 3747.5 | 17 | 4571.9 | 13 | 328.9 | 21 | 328.9 | 4463 |
| CCA-SSG | 262.4 | 17 | 1808.1 | 152 | 2220.2 | 352 | 98.8 | 78 | 98.8 | 340 |
| BGRL | 658.4 | 25 | 3749.8 | 163 | 4574.2 | 373 | 331.2 | 180 | 331.2 | 787 |
| SUGRL | 193.8 | 13 | 2131.2 | 147 | 2615.1 | 342 | 99.5 | 26 | 99.5 | 117 |
| GGD | 254.7 | 15 | 3747.3 | 140 | 4571.6 | 340 | 30.0 | 100 | 30.0 | 1400 |
| Average | 490.4 | 11.9 | 3198.7 | 104.5 | 3906.0 | 237.6 | 246.4 | 68.3 | 246.4 | 1279.1 |
| SpikeGCL | 60.9 | 0.038 | 460.7 | 0.048 | 564.4 | 0.068 | 7.3 | 0.2 | 6.6 | 0.18 |

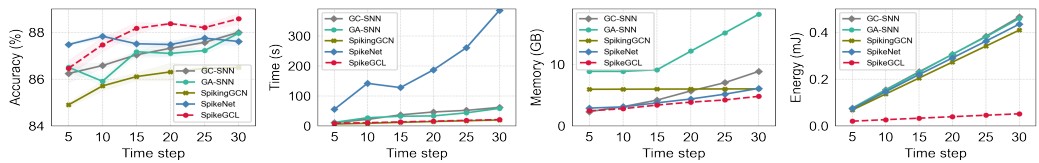

Figure 4: Comparison of SPIKEGCL and other graph SNNs in terms of accuracy (%), training time (s), memory usage (GB), and energy consumption (mJ), respectively.

consumption follows existing works Zhu et al. (2022); Zhou et al. (2022); Wang et al. (2021) and is detailed in Appendix E. Table 2 summarizes the results in terms of model size and energy efficiency compared to full-precision GCL methods. SPIKEGCL consistently shows better efficiency across all datasets, achieving ∼7x less energy consumption and ∼1/60 model size on MAG. As the time step $T$ is a critical hyperparameter for SNNs to better approximate real-valued inputs with discrete spikes, we compare the efficiency of SPIKEGCL and other graph SNNs in terms of accuracy, training time, GPU memory usage, and theoretical energy consumption with varying time steps from 5 to 30. The results on the Computers dataset are shown in Figure 4. It is observed that SPIKEGCL and other graph SNNs benefit from increasing time steps, generally improving accuracy as the time step increases. However, a large time step often leads to more overheads. In our experiments, a larger time step can make SNNs inefficient and become a major bottleneck to scaling to large graphs, as demonstrated by increasing training time, memory usage, and energy consumption. Nevertheless, the efficiency of SPIKEGCL is less affected by increasing time steps. The results demonstrate that SPIKEGCL, coupled with compact graph sequence input and blockwise training paradigm, alleviates such a drawback of training deep SNNs. Overall, SPIKEGCL is able to significantly reduces computation costs and enhance the representational learning capabilities of the model simultaneously.

## 7 CONCLUSION

In this work, we present SPIKEGCL, a principled graph contrastive learning framework to learn binarized and compact representations for graphs at scale. SPIKEGCL leverages the sparse and binary characteristics of SNNs, as well as contrastive learning paradigms, to meet the challenges of increasing graph scale and limited label annotations. The binarized representations learned by SPIKEGCL require less memory and computations compared to traditional GCL, which leads to potential benefits for energy-efficient computing. We provide theoretical guarantees and empirical results to demonstrate that SPIKEGCL is an efficient yet effective approach for learning binarized graph representations. In our extensive experimental evaluation, SPIKEGCL is able to achieve performance on par with advanced baselines using full-precision or 1-bit representations while demonstrating significant efficiency advantages in terms of parameters, speed, memory usage, and energy consumption. We believe that our work is promising from a neuroscientific standpoint, and we hope it will inspire further research toward efficient graph representation learning.

ACKNOWLEDGEMENT

The research is supported by the National Key R&D Program of China under grant No. 2022YFF0902500, the Guangdong Basic and Applied Basic Research Foundation, China (No. 2023A1515011050), Shenzhen Research Project (KJZD20231023094501003), Ant Group Research Intern Program.

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

# A  THEORY AND PROOF

To establish the expressiveness of the proposed model, we define an oracle GNN which performs standard message passing operations with input node feature being the raw features, with per-layer hidden dimension $\frac{d}{T}$. For simplicity, we assume the oracle GNN is instantiated by stacking $L$ GCN layers, with parameters $\theta_* = \{\mathbf{W}_*^{(1)}, \ldots, \mathbf{W}_*^{(L)}\}$. We additionally make two assumptions that are usually satisfied in graph modeling scenarios:

**Assumption 1.** *The input graph $\mathcal{G}$ has its degree bounded from above by $D$.*

**Assumption 2.** *The operator norm of all the weight matrices of the oracle GNN model is bounded from above by $\nu$, i.e., $\sup_{l \in [L]} \|\mathbf{W}_*^{(l)}\|_{op} \leq \nu$*

**Assumption 3.** *The membrane potential at any time step is lower bounded by $V_{lb} < 0$ with $|V_{lb}| \leq \kappa V_{th}$, with some $\kappa > 1$.*

**Remark 1.** *Assumption 3 is actually not necessary and is stated primarily for a sleeker presentation of results. In particular, given assumption 2, we are able to derive a lower bound that depends on the GNN matrices' operator norms, length $T$ as well as the characteristics of the underlying GNN.*

For some node $v$, denote $\widehat{z}_v = \frac{1}{T} \sum_{t=1}^{T} s_v$ as the *firing rate* derived from the binaried representation learned by SNNs. The following theorem states that with a sufficiently large time window $T$, the proposed SNN can implement the oracle GNN with a small approximation error.

**Theorem 2.** *Assume that assumptions 1-3 hold, for any input graph $\mathcal{G}$, let $z_v^*$ be the representation of node $v$ produced by some $L$-layer oracle GCN model. Under the IF model and reset by subtraction mechanism, there exists an SNN such that its firing rate $\widehat{z}_v$ has the following approximation property:*

$$\sup_{\mathcal{G}} \sup_{v \in \mathcal{V}} \|\widehat{z}_v - z_v^*\|_2 \leq \frac{\sqrt{d}\kappa \left(\sqrt{1 + D}\nu\right)^L}{T\sqrt{T}} \tag{10}$$

With a slight abuse of notation, hereafter we will denote $V_{(\cdot)}$ as either a scalar with a value equal to $V_{(\cdot)}$ or a vector of length $\frac{d}{T}$ with each coordinate being $V_{(\cdot)}$, the meaning shall be clear from the context. The proof relies on the following simple Lemma:

**Lemma 1.** *For any node $v$ and $1 \leq l \leq L$, let $V_v^l(t)$ and $H_v^l(t)$ be the membrane potential and input current at time step $1 \leq t \leq T$ and layer $l$, further let $N_v^l(T) = \sum_{t=1}^{T} s_v(t)$ be the number of fired spikes throughout the $T$ steps. We have the following inequality:*

$$V_v^l(T) = \sum_{t=1}^{T} H_v^l(t) - N_v^l(T)V_{th} \tag{11}$$

*Proof of Lemma 1.* The Lemma follows from the trivial fact that each time upon firing, the membrane potential reduces by exactly $V_{th}$. □

*Proof of theorem 2.* We will first give the construction of the approximation SNN. Recall that the parameter configuration of an $L$-layer SNN is given by:

**1st-layer** Since there are $T$ distinct GNN encoders, denote the corresponding weight matrices as $\mathbf{W}_1^{(1)}, \ldots, \mathbf{W}_T^{(1)}$.

**2nd to $L$th layer (if exists)** We equip each layer with a single weight matrix $\mathbf{W}^{(l)}, 2 \leq l \leq L$.

Now we construct the approximation SNN as follows: Given the oracle GNN with parameter $\theta_* = \{\mathbf{W}_*^{(1)}, \ldots, \mathbf{W}_*^{(L)}\}$. In the first layer, we partition $\mathbf{W}_*^{(1)}$ row-wise into $T$ blocks, with each block having an identical number of rows if $d$ is divided by $T$, otherwise the first $T - 1$ blocks constructed with row $d//T$ with the last one having $d \bmod T$ rows, denote the resulting blocks $\mathbf{W}_{*,1}^{(1)}, \ldots, \mathbf{W}_{*,T}^{(1)}$. Then we set $\mathbf{W}_t^{(1)} = T\mathbf{W}_{*,t}^{(1)}, 1 \leq t \leq T$. In the subsequent layers, we set $\mathbf{W}^{(l)} = V_{th}\mathbf{W}_*^{(l)}$. For notational simplicity, we define $\alpha_{uv} = \frac{1}{\sqrt{(|\mathcal{N}_u|+1)\cdot(|\mathcal{N}_v|+1)}}$ for some node pair

$(u, v)$. Now fix some arbitrary node $v \in V$. Recall the aggregation equation at the $l$-th layer of the SNN, with time step $t$:

$$H_v^l(t) = \begin{cases} \sum_{u \in \mathcal{N}_v \cup \{v\}} a_{uv} \mathbf{W}^{(l)} s_v^{l-1}(t), & \text{if } 2 \leq l \leq L \\ \sum_{u \in \mathcal{N}_v \cup \{v\}} a_{uv} \mathbf{W}_t^{(1)} x_v^t, & \text{if } l = 1 \end{cases}. \tag{12}$$

Here note that in the first layer, the pre-activation output of the oracle GNN is $\frac{1}{T} \sum_{t=1}^T H_v^1(t) = \frac{1}{T} \sum_{t=1}^T s_v^1(t)$ for any node $v$. Now we average Eq. 12 at the $L$-th (final) layer according to all timesteps, yielding

$$\frac{1}{T} \sum_{t=1}^T H_v^L(t) = \frac{1}{T} \sum_{u \in \mathcal{N}_v \cup \{v\}} a_{uv} V_{\text{th}} \mathbf{W}_*^{(L)} \sum_{t=1}^T s_u^{L-1}(t) \tag{13}$$

$$= \sum_{u \in \mathcal{N}_v \cup \{v\}} a_{uv} V_{\text{th}} \mathbf{W}_*^{(L)} \widehat{z}_v^{L-1}. \tag{14}$$

Now use Lemma 1 at layer $L$:

$$\widehat{z}_v = \widehat{z}_v^L = \frac{N_v^L(T)}{T} = \frac{1}{V_{\text{th}}} \left( \frac{1}{T} \sum_{t=1}^T H_v^L(t) - \frac{V_v^L(T)}{T} \right) \tag{15}$$

$$= \sum_{u \in \mathcal{N}_v \cup \{v\}} a_{uv} \mathbf{W}_*^{(L)} \widehat{z}_u^{L-1} - \frac{V_v^L(T)}{T V_{\text{th}}} \tag{16}$$

$$= \underbrace{\sum_{u \in \mathcal{N}_v \cup \{v\}} a_{uv} \mathbf{W}_*^{(L)} \text{ReLU} \left( \widehat{z}_u^{L-1} \right) - \frac{V_v^L(T)}{T V_{\text{th}}}}_{\mathcal{T}_1}, \tag{17}$$

where the last equality follows since the firing rates are by definition non-negative. Next we further analyze $\mathcal{T}$, apply Lemma 1 at layer $L-1$ yields:

$$\mathcal{T}_1 = \sum_{u \in \mathcal{N}_v \cup \{v\}} a_{uv} \mathbf{W}_*^{(L)} \text{ReLU} \left( \sum_{\omega \in \mathcal{N}_u \cup \{u\}} a_{u\omega} \mathbf{W}_*^{(L-1)} \widehat{z}_\omega^{L-2} - \frac{V_u^{L-1}(T)}{T V_{\text{th}}} \right) \tag{18}$$

$$= \sum_{u \in \mathcal{N}_v \cup \{v\}} \left[ a_{uv} \mathbf{W}_*^{(L)} \text{ReLU} \left( \sum_{\omega \in \mathcal{N}_u \cup \{u\}} a_{u\omega} \mathbf{W}_*^{(L-1)} \widehat{z}_\omega^{L-2} \right) + a_{uv} \mathbf{W}_*^{(L)} \Upsilon^{L-1} \right] \tag{19}$$

$$= \sum_{u \in \mathcal{N}_v \cup \{v\}} a_{uv} \mathbf{W}_*^{(L)} \text{ReLU} \left( \sum_{\omega \in \mathcal{N}_u \cup \{u\}} a_{u\omega} \mathbf{W}_*^{(L-1)} \text{ReLU} \left( \widehat{z}_\omega^{L-2} \right) \right) + \underbrace{\sum_{u \in \mathcal{N}_v \cup \{v\}} a_{uv} \mathbf{W}_*^{(L)} \Upsilon^{L-1}}_{\mathcal{R}^{L-1}} \tag{20}$$

where in the remainder term $\mathcal{R}^{L-1}$, we define

$$\Upsilon^{L-1} = \text{ReLU} \left( \sum_{\omega \in \mathcal{N}_u \cup \{u\}} a_{u\omega} \mathbf{W}_*^{(L-1)} \widehat{z}_\omega^{L-2} - \frac{V_u^{L-1}(T)}{T V_{\text{th}}} \right) - \text{ReLU} \left( \sum_{\omega \in \mathcal{N}_u \cup \{u\}} a_{u\omega} \mathbf{W}_*^{(L-1)} \widehat{z}_\omega^{L-2} \right) \tag{21}$$

We will first obtain an upper bound of the $\ell_2$-norm of $\Upsilon^{L-1}$, specifically, using the inequality $|\text{ReLU}(x+y) - \text{ReLU}(x)| \leq |y|$,

$$\left\| \Upsilon^{L-1} \right\|_2 \leq \left\| \frac{V_u^{L-1}(T)}{T V_{\text{th}}} \right\|_2 = \frac{\left\| V_u^{L-1}(T) \right\|_2}{T V_{\text{th}}} \leq \sqrt{\frac{d}{T}} \frac{\kappa}{T} \tag{22}$$

where the last inequality follows from the fact the membrane potential never exceeds $V_{\text{th}}$ and assumption 3. Next we bound the term $\mathcal{R}^{L-1}$ under the GCN model, we further define $\deg(v) = |\mathcal{N}_v|$. We have:

$$\left\|\mathcal{R}^{L-1}\right\|_2 \leq \sum_{u \in \mathcal{N}_v \cup \{v\}} \frac{1}{\sqrt{(1+\deg(u))(1+\deg(v))}} \left\|\mathbf{W}_*^{(L)} \Upsilon^{L-1}\right\|_2 \tag{23}$$

$$\leq \sum_{u \in \mathcal{N}_v \cup \{v\}} \frac{1}{\sqrt{(1+\deg(u))(1+\deg(v))}} \nu \sqrt{\frac{d}{T}} \frac{\kappa}{T} \tag{24}$$

$$\leq \sqrt{1+D}\nu \sqrt{\frac{d}{T}} \frac{\kappa}{T}, \tag{25}$$

where the second inequality follows by assumption 2, and the last inequality follows by the bounded degree assumption 1 and $\deg(u) \geq 0$. Now we further unravel the right hand side of Eq. 20 untill there are altogether $L-1$ remainders. It is straightforward to check that the leftmost term would be $z_v^*$, combining with Eq. 17, we write

$$\widehat{z}_v = z_v^* + \mathcal{R}^1 + \cdots + \mathcal{R}^{L-1} + \mathcal{R}^L, \tag{26}$$

with $\mathcal{R}^L = -\frac{V_v^L(T)}{TV_{\text{th}}}$ being the remainder term in Eq. 17. Following similar arguments, we can show that the remainder terms satisfy:

$$\left\|\mathcal{R}^l\right\|_2 \leq \left(\sqrt{1+D}\nu\right)^{L-l} \sqrt{\frac{d}{T}} \frac{\kappa}{T}, \qquad 1 \leq l \leq L, \tag{27}$$

Consequently we have:

$$\left\|\widehat{z}_v - z_v^*\right\|_2 \leq \left\|\mathcal{R}^1\right\|_2 + \cdots + \left\|\mathcal{R}^L\right\|_2 \tag{28}$$

$$\leq \sqrt{\frac{d}{T}} \frac{\kappa}{T} \left(1 + \cdots + \left(\sqrt{1+D}\nu\right)^{L-1}\right) \tag{29}$$

$$\leq \frac{\sqrt{d}\kappa \left(\sqrt{1+D}\nu\right)^L}{\sqrt{T}T} \tag{30}$$

$\square$

**Remark 2.** *The approximation bound Eq. 10 is dependent on the specific form of GCN, and therefore has the scaling term $O\left((\nu\sqrt{1+D})^L\right)$ which is exponential in $L$. With an alternative underlying GNN model, we may get better approximations, for example in the SAGE model Hamilton et al. (2017), following similar arguments, we may prove that we may remove the exponential dependence over (root)-degree bounds, i.e., the scaling factor reduces to $O\left(L\nu^L\right)$.*

**Remark 3.** *Theorem 2 relies on the reset by subtraction mechanism, for the reset to constant (zero) mechanism, as stated in Rueckauer et al. (2016), we have to sacrifice an error term that does not vanish as $T$ grows even in the i.i.d setting.*

## B  VANISHING GRADIENTS IN SURROGATE LEARNING

The vanishing gradient problem is a well-known issue that hinders convergence in modern neural networks. To address this problem, ReLU activation functions were introduced as alternatives to Sigmoid and Tanh to avoid dead neurons during backpropagation. However, in SNNs, the vanishing gradient problem can arise again when using a layer of spiking neurons to learn long sequences, similar to the vanishing problem in deep artificial neural networks. We attribute the vanishing gradient problem in SNNs to the surrogate gradient learning technique, wherein surrogate functions (typically smooth Sigmoid or Tanh) are used to approximate the gradient of the Heaviside step function in spiking neurons. While surrogate learning plays a crucial role in directly training SNNs, it inherits the defects of surrogate functions, resulting in poor training performance and slow convergence. Recent works, such as Fang et al. (2021); Feng et al. (2022), have also noted this issue in SNNs and reached the same conclusion.

Addressing the vanishing gradient problem is essential for unlocking the full potential of deep SNNs and enabling them to learn and generalize effectively. Recent success in local learning Siddiqui et al. (2023) has pointed out a new way to train deep modern networks without suffering severe network degradation and vanishing gradients problems. Specifically, the network is divided into several individual blocks that are trained locally with gradient isolation during training to avoid a long backpropagation path. Inspired by this approach, we are motivated to explore an interesting research direction in SNNs: *Is it possible to address the issue of vanishing gradients by adjusting the backpropagation path in a reasonable way?*

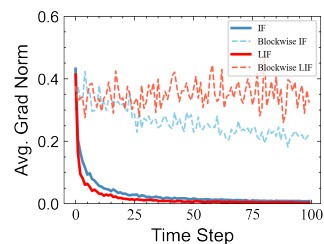

Figure 5: Average gradient norms of IF and LIF neurons, with and without the blockwise learning strategy.

In this work, we present a blockwise training strategy to address the vanishing gradient problem in SNNs. Unlike conventional end-to-end learning, we horizontally divide the encoder network into several individual blocks along the time dimension[2]. We limit the length of the backpropagation path to each of these blocks and apply a self-supervised contrastive loss locally at different blocks, using *stop-gradient* to ensure that the loss function from one block does not affect the learned parameters of other blocks. We argue that the blockwise learning rule is unlikely to cause vanishing gradients as long as each block has a reasonable depth. We plot the averaged gradient norm of IF and LIF in Figure 5 and observe that signals in both IF and LIF neurons are prone to vanishing with an increasing time step. However, the backward gradients exhibit healthy norms in IF and LIF, and the vanishing gradient problem is largely alleviated by incorporating our blockwise optimization paradigm.

## C  ALGORITHM

---

**Algorithm 1** Spiking Graph Contrastive Learning (SPIKEGCL)

---

**Input:** Graph $\mathcal{G} = (\mathcal{V}, \mathcal{E}, \mathbf{X})$, encoder $f_\theta(\cdot) = [f_{\theta_1}^1, \ldots, f_{\theta_T}^T]$, predictor head $g_\phi(\cdot)$, time step $T$;
**Output:** Learned encoder $f_\theta(\cdot)$;

1: **while** *not converged* **do**;
2:      $\tilde{\mathcal{G}} = (\mathcal{V}, \mathcal{E}, \tilde{\mathbf{X}}) \leftarrow$ corruption$(\mathcal{G})$;
3:      **for** $t = 1$ to $T$ **do**
4:          $\mathbf{Z_1} \leftarrow f_{\theta_t}^t(\mathcal{G}^t)$;
5:          $\mathbf{H_1} \leftarrow g_\phi(\mathbf{Z_1})$;
6:          $\mathbf{Z_2} \leftarrow f_{\theta_t}^t(\tilde{\mathcal{G}}^t)$;
7:          $\mathbf{H_2} \leftarrow g_\phi(\mathbf{Z_2})$;
8:          Calculate contrastive loss $\mathcal{J}$ over $\mathbf{H_1}, \mathbf{H_2}$;
9:          Update $\theta_t, \phi$ by gradient descent;
10:     **end for**
11: **end while**;
12: **return** $f_\theta(\cdot)$;

---

---

[2]Our work differs from Siddiqui et al. (2023), in which the network is vertically divided from bottom to top.

**Algorithm 2** PyTorch-style pseudocode for SPIKEGCL

```
# f: a set of peer GNNs
# h: SNN neuron
# g: MLP network
# m: margin
# perm: random permutation of node features
# T: time steps
# G: input graph
# x: node features

x1 = torch.chunk(x, T, dim=1)
G1 = G
# corruption
x2 = torch.chunk(x[:, perm)], T, dim=1)
G2 = drop_edge(G)
for t in range(T):
    z1 = f[t](x1[t], G1)
    # gradient isolation
    z1 = h(z1.detach())
    z1 = g(z1)

    z2 = f(x2[t], G2)
    # gradient isolation
    z2 = h(z2.detach())
    z2 = g(z2)
    # compute loss & backpropagation
    loss = F.margin_ranking_loss(z1, z2, margin=m)
    loss.backward()
```

# D DISCUSSION

## D.1 DISCUSSION ON COMPLEXITY

As SPIKEGCL uses a relatively simple predictor head (a single-layer MLP), the main complexity bottleneck lies in the GNN-based encoder network. Therefore, we discuss the overall computation and parameter complexity of SPIKEGCL from the perspective of the encoder. Recall that our encoder consists of $T$ peer GNNs coupled with a spiking neuron. The computation and parameter complexity of our method is generally $T$ times larger than standard GCL methods that use a single GNN as an encoder. Here we omit the bias term in GNN and learnable parameters (if have) in spiking neurons as they do not significantly affect the computation and parameter complexity. Note that we divide the input features into $T$ groups and share parameters across different GNNs. Each GNN actually has only $\frac{1}{T}$ size of parameters with a hidden dimension of $\frac{d}{T}$ compared to the standard case. Therefore, our method achieves similar computation and parameter complexity as traditional GCL methods that use a single GNN as an encoder. Since each GNN accepts each group of features as input and performs message aggregation individually, the computation can be trivially parallelized, which accelerates computations on larger graphs. This further reduces the complexity of our method and endows it with desirable scalability not only to larger dataset sizes but also to larger embedding dimensions.

## D.2 LIMITATION

We note certain limitations of our work. (i) First, the linear evaluation used in our experiments assumes that the downstream task can be solved by a linear classifier, which may not always be the case. In addition, the quality of the learned graph representations may depend on the specific downstream task of interest, and linear evaluation may not be able to capture all relevant aspects of the learned representations for the task. (ii) Second, one common assumption behind the theoretical guarantees is that the neuron adopts a 'reset by subtraction' mechanism, which may not always hold

for spiking neurons that use the 'reset to zero' mechanism, as also mentioned in Remark 3. (iii) Third, our empirical evaluations on real-world datasets do not include any significantly large-scale datasets ($N \sim 10^6$ or higher), although our collection of 9 datasets is a broad selection among those commonly used in related research. These aspects mark potential areas for future enhancements and investigations

# E    DETAILED EXPERIMENTAL SETTINGS

Table 3: Dataset statistics.

|  | Computers | Photo | CS | Physics | Cora | CiteSeer | PubMed | arXiv | MAG |
|---|---|---|---|---|---|---|---|---|---|
| **#Nodes** | 13,752 | 7,650 | 18,333 | 34,493 | 2,708 | 3,327 | 19,717 | 16,9343 | 736,389 |
| **#Edges** | 491,722 | 238,162 | 163,788 | 495,924 | 10,556 | 9,104 | 88,648 | 2,315,598 | 10,792,672 |
| **#Features** | 767 | 745 | 6,805 | 8,415 | 1,433 | 3,703 | 500 | 128 | 128 |
| **#Classes** | 10 | 8 | 15 | 5 | 7 | 6 | 3 | 40 | 349 |
| **Density** | 0.144% | 0.082% | 0.023% | 0.407% | 0.260% | 0.049% | 0.042% | 0.008% | 0.002% |

**Datasets.** We evaluate SPIKEGCL on several graph benchmarks with different scales and properties. Following prior works Zhang et al. (2021); Thakoor et al. (2021); Zhu et al. (2022), we adopt 9 common benchmark graphs, including two co-purchase graphs, i.e., Amazon-Photo, Amazon-Computer Shchur et al. (2018), two co-author graphs, i.e., Coauthor-CS and Coauthor-Physics Shchur et al. (2018), three citation graphs, i.e., Cora, CiteSeer, PubMed Sen et al. (2008), as well as two large-scale datasets ogbn-arXiv and ogbn-MAG from Open Graph Benchmark Hu et al. (2020). Dataset statistics are listed in Table 3. For three citation datasets, we evaluate the models on the public *full* splits introduced in Rong et al. (2020); Chen et al. (2018). we adopt the random 1:1:8 split for Amazon-Computer, Amazon-Photo, Coauthor-CS, and Coauthor-Physics. We use stratified sampling to ensure that the class distribution remains the same across splits.

**Baselines.**    We compare our proposed methods to a wide range of baselines that fall into four categories:

- **Full-precision (supervised) GNNs**: GCN Kipf & Welling (2016) and GAT Veličković et al. (2018).

- **1-bit quantization-based GNNs**: Bi-GCN Wang et al. (2021), BinaryGNN Bahri et al. (2021) and BANE Yang et al. (2018). Bi-GCN and BinaryGNN are both supervised methods while BANE is an unsupervised one.

- **Full-precision contrastive methods**: DGI Velickovic et al. (2019), GRACE Zhu et al. (2020), CCA-SSG Zhang et al. (2021), BGRL Thakoor et al. (2021), SUGRL Mo et al. (2022), and GGD Zheng et al. (2022). DGI, GRACE, SUGRL, and GGD are negative-sampling-based methods, whereas BGRL and CCA-SSG are negative-sample-free ones.

- **Graph SNNs**: SpikingGCN Zhu et al. (2022), SpikeNet Li et al. (2023b), GC-SNN and GA-SNN Xu et al. (2021). Note that SpikeNet is initially designed for dynamic graphs, we adapt the author's implementation to static graphs following the practice in Zhu et al. (2022); Xu et al. (2021). Although these methods are also based on the combination of SNNs and GNNs, they do not fully utilize the binary nature of SNNs to learn binaried representations in an unsupervised fashion.

The hyperparameters of all the baselines were configured according to the experimental settings officially reported by the authors and were then carefully tuned in our experiments to achieve their best results.

**Implementation details.**    We present implementation details for our experiments for reproducibility. We implement our model as well as the baselines with PyTorch Paszke et al. (2019) and PyTorch Geometric Fey & Lenssen (2019), which are open-source software released under BSD-style [3] and MIT[4] license, respectively. All datasets used throughout experiments are publicly available in

---

[3] https://github.com/pytorch/pytorch/blob/master/LICENSE
[4] https://github.com/pyg-team/pytorch_geometric/blob/master/LICENSE

PyTorch Geometric library. All experiments are conducted on an NVIDIA RTX 3090 Ti GPU with 24 GB memory unless specified. Code is available at `https://github.com/EdisonLeeeee/SpikeGCL` for reproducibility.

**Hyperparameter settings.** The embedding dimension for each time step is searched in $\{4, 8, 16, 32, 64\}$. We carefully tune the time step $T$ from 8 to 64 to better approximate the full-precision performance. The firing threshold is tuned within $\{5e\text{-}4, 5e\text{-}3, 5e\text{-}2, 5e\text{-}1, 1.0\}$. The margin $m$ is tuned in $\{0, 0.5, 1.0, 1.5, 2.0\}$. We initialize and optimize all models with default normal initializer and AdamW optimizer Loshchilov & Hutter (2019). For all datasets, we follow the practice in Li et al. (2023b) and increase the sparsity of the network using PLIF Fang et al. (2020) units, a bio-inspired neuron model that adopts a learnable $\tau_m = 1$ to activate sparsely in time solely when crossing a threshold. We adopt Sigmoid as the surrogate function during backpropagation, with a smooth factor $\alpha = 2.0$. Exploration on parameters of SPIKEGCL including time step $T$, encoder architecture, spiking neurons, and threshold $V_{\text{th}}$ are elaborated in Appendix F.

**Evaluation protocol.** For unsupervised or self-supervised methods, including SPIKEGCL, we follow the *linear evaluation* scheme as described in literature Velickovic et al. (2019); Thakoor et al. (2021): (i) we first train the model on all the nodes in a graph with self-defined supervision to learn node representations; (ii) we then train a linear classifier (e.g., a logistic regression model) on top with the learned representations under the supervised setting to evaluate the performance. We report averaged accuracy with standard deviation across 10 different trials with random seeds.

**Computation of theoretical energy consumption.** Different from common ANNs implemented on GPUs, SNNs are designed for neuromorphic chips that adopt synaptic operations (SOPs) to run neural networks in a low power consumption manner. However, training SNNs directly on neuromorphic chips has been rarely explored in the literature. To investigate the energy consumption of SNN-based methods on neuromorphic chips, we follow previous works Zhu et al. (2022); Zhou et al. (2022) and adopt an alternative estimation approach, which involves counting the total number of spikes generated during the embedding generation process. The computation of energy consumption involves two parts. The first one comes from the spike encoding process, which converts the full-precision node representations into discrete spikes to run on neuromorphic chips (e.g., Bernoulli encoding). It is estimated by multiply-and-accumulate (MAC) operations. The second one is the spiking process, which receives the resultant current and accumulates the membrane potential to generate the spikes. Therefore, the theoretical energy consumption can be estimated as follows:

$$
\begin{aligned}
E &= E_{\text{encoding}} + E_{\text{spiking}} \\
&= E_{\text{MAC}} \sum_{t=1}^{T} Nd + E_{\text{SOP}} \sum_{t=1}^{T} \sum_{l=1}^{L} \mathbf{S}_t^l,
\end{aligned}
\tag{31}
$$

where $\mathbf{S}_t^l$ denotes the output spikes at time step $t$ and layer $l$. Note that SPIKEGCL has only one spiking neuron, so we count the number of spikes at the last layer. According to Zhu et al. (2022); Zhou et al. (2022), $E_{\text{MAC}}$ and $E_{\text{SOP}}$ are set to $4.6pJ$ and $3.7pJ$, respectively. We also detail how to calculate the theoretical energy consumption of methods other than graph SNNs. For 1-bit GNNs, the processing time required to execute a single cycle operation, encompassing one multiplication and one addition, can be utilized to perform 64 binary operations effectively Rastegari et al. (2016). Therefore, we use the following formula derived from Wang et al. (2021) to calculate the theoretical energy consumption of 1-bit GNNs (specifically GCN):

$$
E = E_{\text{MAC}}(\frac{1}{64} Nd_{in}^l d_{out}^l + 2Nd_{out}^l + |\mathcal{E}|d_{out}^l),
\tag{32}
$$

where $d_{in}^l$ and $d_{out}^l$ are the input and output dimensions of the representation at the $l$-th layer. We assume $d_{in}^l = d_{out}^l = d$ in this paper. For the other full-precision models deployed on GPUs, we count the number of MAC operations to calculate the theoretical energy consumption. In practice, we split the message passing into two steps: embedding generation and aggregation. In the embedding generation step, the vanilla GNN projects features into a low-dimensional embedding space. For example, given the node features $\mathbf{X}$ and a weight matrix $\mathbf{W} \in \mathbb{R}^{d_{in} \times d_{out}}$, it executes $Nd_{in}d_{out}$ multiplication and $Nd_{in}d_{out}$ addition operations. In the aggregation step, the number of multiplication and addition operations can be simply considered as $|E|d_{in}$ and $|E|d_{out}$. For a fair comparison, we only consider the energy consumption of the encoder in GCL, as it is often the main bottleneck.

Table 4: Classification accuracy (%) on three citation datasets. The best result for each dataset is highlighted in **red**. Darker colors indicate larger performance gaps between SPIKEGCL and the best results. (**U**: unsupervised or self-supervised; **S**: spike-based; **B**: binarized)

| Methods | U | S | B | Cora | CiteSeer | PubMed |
|---|---|---|---|---|---|---|
| GCN Kipf & Welling (2016) | | | | $86.1_{\pm 0.2}$ | $75.9_{\pm 0.4}$ | $88.2_{\pm 0.5}$ |
| GAT Veličković et al. (2018) | | | | $86.7_{\pm 0.7}$ | $78.5_{\pm 0.4}$ | $86.8_{\pm 0.3}$ |
| GC-SNN Xu et al. (2021) | | ✓ | | $83.8_{\pm 0.4}$ | $73.4_{\pm 0.5}$ | $85.3_{\pm 0.4}$ |
| GA-SNN Xu et al. (2021) | | ✓ | | $86.4_{\pm 0.2}$ | $72.8_{\pm 0.8}$ | $84.5_{\pm 0.6}$ |
| SpikeNet Li et al. (2023b) | | ✓ | | $83.5_{\pm 0.3}$ | $71.4_{\pm 0.5}$ | $83.9_{\pm 0.4}$ |
| SpikingGCN Zhu et al. (2022) | | ✓ | | $87.7_{\pm 0.4}$ | $77.7_{\pm 0.3}$ | $87.5_{\pm 0.5}$ |
| Bi-GCN Wang et al. (2021) | | | ✓ | $85.1_{\pm 0.6}$ | $72.8_{\pm 0.7}$ | $88.6_{\pm 1.0}$ |
| Binary GNN Bahri et al. (2021) | | | ✓ | $85.0_{\pm 0.9}$ | $71.9_{\pm 0.2}$ | $86.8_{\pm 0.7}$ |
| BANE Yang et al. (2018) | ✓ | | ✓ | $65.0_{\pm 0.4}$ | $64.3_{\pm 0.3}$ | $68.8_{\pm 0.2}$ |
| DGI Velickovic et al. (2019) | ✓ | | | $86.3_{\pm 0.2}$ | $\textbf{78.9}_{\pm 0.2}$ | $86.2_{\pm 0.1}$ |
| GRACE Zhu et al. (2020) | ✓ | | | $87.2_{\pm 0.2}$ | $74.5_{\pm 0.1}$ | $87.3_{\pm 0.1}$ |
| CCA-SSG Zhang et al. (2021) | ✓ | | | $84.6_{\pm 0.7}$ | $75.4_{\pm 1.0}$ | $88.4_{\pm 0.6}$ |
| BGRL Thakoor et al. (2021) | ✓ | | | $87.3_{\pm 0.1}$ | $76.0_{\pm 0.2}$ | $88.3_{\pm 0.1}$ |
| SUGRL Mo et al. (2022) | ✓ | | | $\textbf{88.0}_{\pm 0.1}$ | $77.6_{\pm 0.4}$ | $88.2_{\pm 0.2}$ |
| GGD Zheng et al. (2022) | ✓ | | | $86.2_{\pm 0.2}$ | $75.5_{\pm 0.1}$ | $84.2_{\pm 0.1}$ |
| SPIKEGCL | ✓ | ✓ | ✓ | $87.4_{\pm 0.6}$ | $77.6_{\pm 0.6}$ | $\textbf{88.8}_{\pm 0.3}$ |

Table 5: The parameter size (KB) and theoretical energy consumption (mJ) of various methods.

| | Computers | | Photo | | CS | | Physics | | arXiv | | MAG | |
|---|---|---|---|---|---|---|---|---|---|---|---|---|
| | #Param↓ | Energy↓ | #Param↓ | Energy↓ | #Param↓ | Energy↓ | #Param↓ | Energy↓ | #Param↓ | Energy↓ | #Param↓ | Energy↓ |
| GC-SNN | 195.9 | 0.53 | 258.9 | 0.29 | 1812.2 | 0.16 | 2221.6 | 0.11 | - | - | - | - |
| GA-SNN | 265.0 | 0.53 | 258.8 | 0.26 | 1812.2 | 0.17 | 1094.4 | 0.11 | - | - | - | - |
| SpikeNet | 293.6 | 0.71 | 484.1 | 0.62 | 1790.2 | 0.46 | 2197.1 | 0.23 | 194.0 | 0.41 | - | - |
| SpikingGCN | 7.6 | 0.68 | 5.9 | 0.36 | 102.0 | 0.20 | 42.0 | 0.21 | - | - | - | - |
| Average † | 190.5 | 0.61 | 251.9 | 0.38 | 1388.7 | 0.24 | 1388.7 | 0.16 | 194.0 | 0.41 | - | - |
| Bi-GCN | 529.9 | 0.67 | 517.6 | 0.33 | 3623.9 | 0.83 | 4443.4 | 2 | 218.1 | 2 | 376.6 | 12.3 |
| BinaryGNN | 108.4 | 1 | 105.3 | 0.57 | 881.9 | 3 | 1086.7 | 11 | 30.5 | 1 | - | - |
| Average † | 319.1 | 0.93 | 311.4 | 0.44 | 2252.9 | 1.86 | 2765.0 | 6.3 | 124.3 | 1.50 | 376.6 | 12.3 |
| DGI | 917.5 | 0.5 | 906.2 | 0.21 | 4008.9 | 8 | 4833.3 | 6 | 590.3 | 5 | 590.3 | 568 |
| GRACE | 656.1 | 1.1 | 644.8 | 0.47 | 3747.5 | 17 | 4571.9 | 13 | 328.9 | 21 | 328.9 | 4463 |
| CCA-SSG | 262.4 | 17 | 256.7 | 9 | 1808.1 | 152 | 2220.2 | 352 | 98.8 | 78 | 98.8 | 340 |
| BGRL | 658.4 | 25 | 647.1 | 13 | 3749.8 | 163 | 4574.2 | 373 | 331.2 | 180 | 331.2 | 787 |
| SUGRL | 193.8 | 13 | 189.4 | 6 | 2131.2 | 147 | 2615.1 | 342 | 99.5 | 26 | 99.5 | 117 |
| GGD | 254.7 | 15 | 249.2 | 8 | 3747.3 | 140 | 4571.6 | 340 | 30.0 | 100 | 30.0 | 1400 |
| Average † | 490.4 | 11.9 | 482.2 | 7 | 3198.7 | 104.5 | 3906.0 | 237.6 | 246.4 | 68.3 | 246.4 | 1279.1 |
| SPIKEGCL | 60.9 | 0.03 | 28.43 | 0.01 | 460.7 | 0.04 | 564.4 | 0.06 | 7.3 | 0.2 | 6.6 | 0.18 |

†Averaged results of baseline methods in each section.

## F ADDITIONAL EMPIRICAL RESULTS

**Performance comparison.** We show additional results of SPIKEGCL and baselines on three common citation benchmarks: Cora, CiteSeer, and PubMed. We can observe from Table 4 that SPIKEGCL obtains competitive performance compared to state-of-the-art baselines across all datasets, including those using full-precision or binarized representations. This highlights that SPIKEGCL is a versatile and effective method for learning node representations in citation networks. It is worth noting that SPIKEGCL achieves these results with significantly fewer bits compared to full-precision representations, which demonstrates its efficiency and scalability. Additionally, SPIKEGCL does not require end-to-end supervision, which makes it more practical for real-world applications.

**Parameter size and energy consumption.** We calculate the parameter size and the theoretical energy consumption of all methods on six different datasets, as shown in Table 5. Note that BANE, a factorization-based approach, is not feasible for estimating its energy consumption, and therefore is omitted from the comparison. Table 5 demonstrates the energy efficiency of graph SNNs and 1-bit GNNs over conventional GCL methods, which have significantly lower energy consumption with an

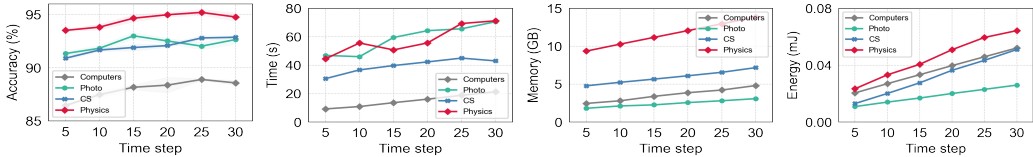

Figure 6: Effect of time step $T$ on Computers, Photo, CS, and Physics datasets.

increasing graph scale. In particular, SPIKEGCL exhibits an average of $\sim$50x, $\sim$100x, and $\sim$1000x less energy consumption compared to graph SNNs, 1-bit GNNs, and GCL baselines, respectively. These results demonstrate that implementing advanced SNNs on energy-efficient hardware can significantly reduce the energy consumption of SPIKEGCL. While we have a series of peer GNNs in our encoders corresponding to each group of inputs, SPIKEGCL also exhibits low complexity in parameter size due to parameter sharing among them. Compared to GCL baselines, the parameter size of SPIKEGCL has decreased by a factor of 8 on average. Additionally, SPIKEGCL has a smaller model size even compared to 1-bit GNNs and graph SNNs. These results demonstrate that SPIKEGCL not only has strong representation learning abilities but also requires fewer parameters compared to other methods.

**Effect of time step $T$.** Figure 6 shows the effect of time step $T$ in terms of accuracy performance, running time, memory usage, and energy consumption. The running time is measured as the training time of SPIKEGCL until converged (with early stopping) and the memory usage refers to the maximum GPU memory usage during the training process. We can observe that as $T$ increases, the accuracy performance of SPIKEGCL gradually improves until it reaches a plateau (e.g., $T = 25$). With the increase of $T$, the running time and memory usage also slightly increase, which may lead to longer training times and higher hardware requirements. Additionally, the energy consumption of SPIKEGCL also increases with larger values of $T$, which can be a concern for energy-efficient applications. Therefore, the choice of $T$ should strike a balance between accuracy performance and computational efficiency, taking into account the specific requirements of the application at hand.

Table 6: Classification accuracy (%) of SPIKEGCL on six datasets with different encoder architectures. The best result for each dataset is highlighted in **red**.

|  | **Computers** | **Photo** | **CS** | **Physics** | **arXiv** | **MAG** |
|---|---|---|---|---|---|---|
| SAGE | $84.4_{\pm 0.9}$ | $91.4_{\pm 0.3}$ | $90.8_{\pm 0.9}$ | $94.6_{\pm 0.1}$ | $65.7_{\pm 0.7}$ | $28.2_{\pm 0.1}$ |
| GAT | $87.9_{\pm 0.6}$ | $92.4_{\pm 0.5}$ | $90.5_{\pm 0.8}$ | $92.6_{\pm 0.3}$ | $68.0_{\pm 0.1}$ | $30.8_{\pm 0.8}$ |
| GCN | $\mathbf{88.9}_{\pm 0.3}$ | $\mathbf{93.0}_{\pm 0.1}$ | $\mathbf{92.8}_{\pm 0.1}$ | $\mathbf{95.2}_{\pm 0.6}$ | $\mathbf{70.9}_{\pm 0.1}$ | $\mathbf{32.0}_{\pm 0.3}$ |

**Effect of encoder architectures.** Different GNN architectures offer SPIKEGCL different capabilities in learning graph structure data. In our experiments, we used GCN as the default architecture in the encoder, our framework allows various choices of GNN architectures though. In order to explore the impact of different GNN architectures on the performance of SPIKEGCL, we conduct ablation studies on three citation graphs with different GNN encoders, including SAGE Hamilton et al. (2017), GAT Veličković et al. (2018), and GCN Kipf & Welling (2016). Table 6 shows that GCN is the most effective architecture for SPIKEGCL, which is consistent with prior works Velickovic et al. (2019); Li et al. (2023a); Thakoor et al. (2021); Zhang et al. (2021). However, it is worth noting that other GNN architectures may still provide valuable insights in different scenarios or for different types of data. Therefore, the choice of GNN architecture should be made based on the specific requirements of the task at hand.

**Effect of spiking neurons.** We provide a series of spiking neurons as building blocks for SNNs, including IF Salinas & Sejnowski (2002), LIF Gerstner et al. (2014), and PLIF Fang et al. (2020). The results are shown in Table 7. It is observed that a simple IF neuron is sufficient for SPIKEGCL to achieve good performance. By introducing a more biologically plausible leaky term, LIF increases the sparsity of output representations and achieves better performance. Additionally, the leaky term can be a learnable parameter, which endows the network with better flexibility and biological plausibility. Therefore, in most cases, PLIF achieves slightly better performance than LIF. Overall, the choice of

Table 7: Classification accuracy (%) of SPIKEGCL on six datasets with different spiking neurons. The best result for each dataset is highlighted in **red**.

|  | **Computers** | **Photo** | **CS** | **Physics** | **arXiv** | **MAG** |
|---|---|---|---|---|---|---|
| IF | $88.1_{\pm 0.6}$ | $92.9_{\pm 0.2}$ | $91.2_{\pm 0.6}$ | $95.0_{\pm 0.5}$ | $68.2_{\pm 0.6}$ | $31.0_{\pm 0.5}$ |
| LIF | $88.6_{\pm 0.1}$ | $92.9_{\pm 0.1}$ | $92.1_{\pm 0.3}$ | $\mathbf{95.3}_{\pm 0.1}$ | $68.7_{\pm 0.8}$ | $29.8_{\pm 0.5}$ |
| PILF | $\mathbf{88.9}_{\pm 0.3}$ | $\mathbf{93.0}_{\pm 0.1}$ | $\mathbf{92.8}_{\pm 0.1}$ | $95.2_{\pm 0.6}$ | $\mathbf{70.9}_{\pm 0.1}$ | $\mathbf{32.0}_{\pm 0.3}$ |

spiking neuron should be based on the specific requirements of the task and the available hardware. For example, IF neurons may be more suitable for tasks with lower computational requirements, while LIF or PLIF neurons may be more suitable for tasks that require higher levels of sparsity or greater biological plausibility.

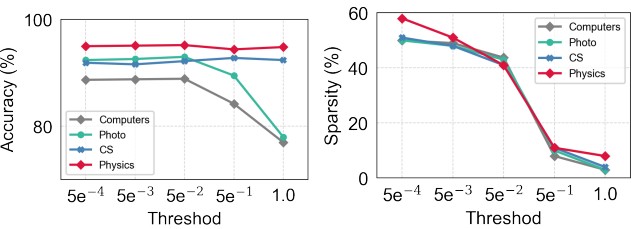

Figure 7: Accuracy (%) and output sparsity (%) of SPIKEGCL in four datasets.

**Effect of threshold $V_{\text{th}}$.** The neuron threshold directly controls the firing rates of spiking neurons and, thereby, the output sparsity. We conducted an ablation study on $V_{\text{th}}$ on four datasets to investigate its effect, as shown in Figure 7. We observed that the performance of SPIKEGCL is sensitive to the value of $V_{\text{th}}$ on two dense datasets, Computers and Photo. When $V_{\text{th}}$ is too low, the output sparsity may be too high, leading to a loss of information and compromised performance. When $V_{\text{th}}$ is too high, the firing rates of the spiking neurons may be too low, resulting in a lack of discriminative power and reduced performance. In contrast, the performance of SPIKEGCL is stable with different $V_{\text{th}}$ on CS and Physics, two relatively sparse datasets. This suggests that a dense dataset requires a smaller $V_{\text{th}}$ to capture the underlying structure. We can also see that the output spikes become sparser as $V_{\text{th}}$ increases. Overall, our ablation study demonstrates the importance of carefully selecting the value of $V_{\text{th}}$ to achieve optimal performance with SPIKEGCL. The optimal value of $V_{\text{th}}$ may depend on the specific dataset; therefore, it should be chosen based on the empirical evaluation.

**Convergence.** We compared the convergence speed of SPIKEGCL with full-precision GCL baselines on Computers, which is shown in Figure 8. It is observed that negative-sample-free methods generally converged faster than negative-sample-based ones. However, they still required sufficient epochs to gradually improve their performance. In contrast, SPIKEGCL that trained in a blockwise training paradigm demonstrates a significantly faster convergence speed over full-precision methods. In particular, 1-2 epochs are sufficient for SPIKEGCL to learn good representations. Overall, our experiments demonstrate the effectiveness of SPIKEGCL in improving the convergence speed of SNNs and provide insights into the benefits of the blockwise training paradigm.

Figure 8: Convergence speed comparison among SPIKEGCL and full-precision methods. Blue: negative-sample-based methods; green: negative-sample-free methods

**Connections between original input features and output spikes.** To gain a deeper understanding of SPIKEGCL, we assess the similarity between input features (partitioned by $T$ groups) and the corresponding output spikes using the Centered Kernel Alignment metric. Centered Kernel Alignment (CKA) Nguyen et al. (2021) serves as a representation similarity metric extensively utilized for comprehending the representations learned by neural networks. CKA takes two representations $\boldsymbol{X}$ and $\boldsymbol{Y}$ as input and calculates their normalized similarity, measured in

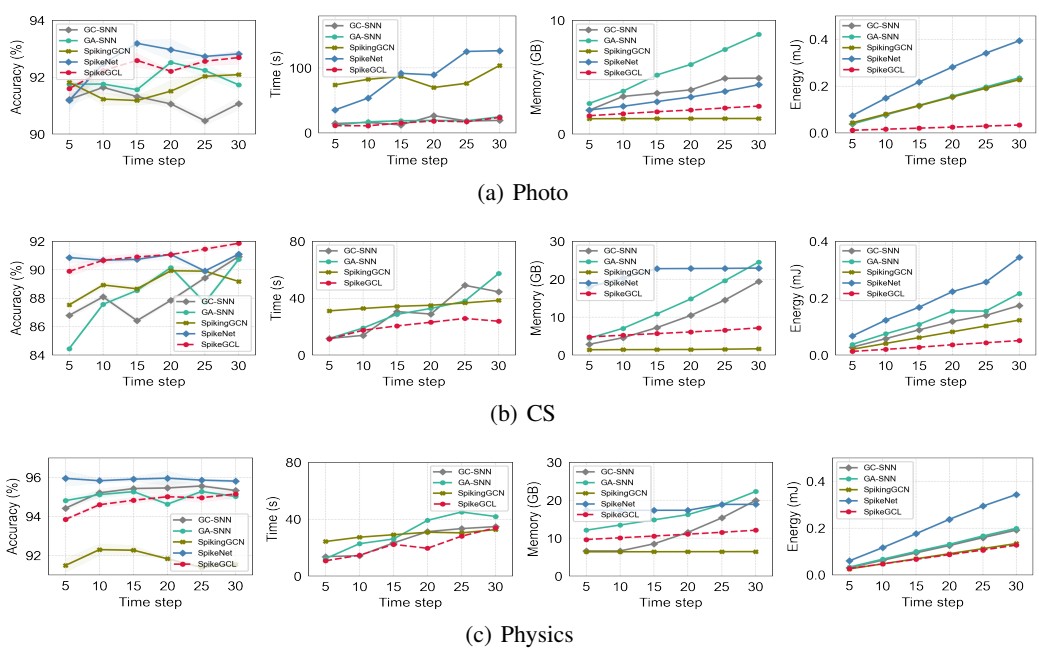

Figure 9: Comparison of SpikeGCL and other graph SNNs in terms of accuracy (%), training time (s), memory usage (GB), and energy consumption (mJ), respectively.

terms of the Hilbert-Schmidt Independence Criterion (HSIC):

$$\text{CKA}(\boldsymbol{K}, \boldsymbol{L}) = \frac{\text{HSIC}_0(\boldsymbol{K}, \boldsymbol{L})}{\sqrt{\text{HSIC}_0(\boldsymbol{K}, \boldsymbol{K}) \text{HSIC}_0(\boldsymbol{L}, \boldsymbol{L})}} \tag{33}$$

Where $\boldsymbol{K}$ and $\boldsymbol{L}$ are similarity matrices of $\boldsymbol{X}$ and $\boldsymbol{Y}$ respectively. The results on Computers and Photo are presented in Figure 10. As evident from the results, each group of features exhibits a strong correlation with the corresponding output spikes while demonstrating minimal correlation with spikes in other time steps. This is reflected in the CKA similarity matrices, with values along the diagonal being notably larger than those elsewhere. The results have also suggested that the learned binary representations are disentangled from each other, thereby providing improved expressiveness in representing the input features.

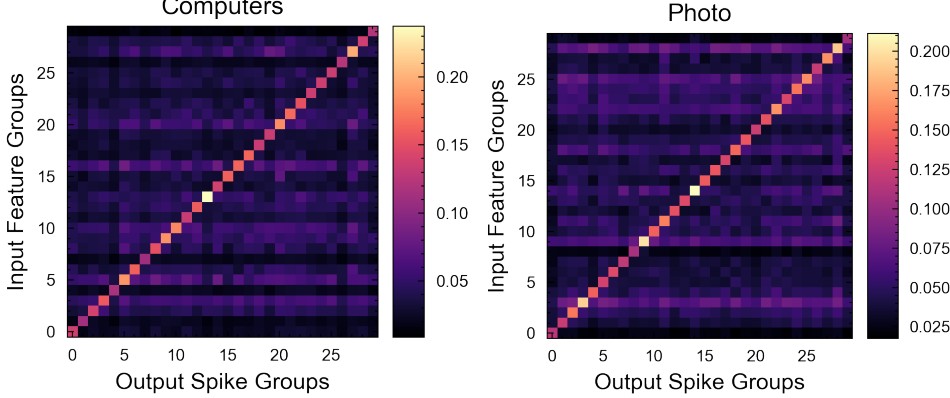

Figure 10: CKA similarity between different input feature groups and output spike groups.

**Effect of reset mechanisms.** We additionally include the ablation results on reset mechanisms of different spiking neurons, as shown in Table 8. As can be observed, the performance of SPIKEGCL

Table 8: Classification accuracy (%) of SPIKEGCL on four datasets with different reset mechanisms and spiking neurons. The best result for each dataset is highlighted in **red**.

|  | Computers | Photo | CS | Physics |
|---|---|---|---|---|
| zero-reset + IF | $88.0_{\pm0.2}$ | $92.7_{\pm0.3}$ | $91.2_{\pm0.1}$ | $95.0_{\pm0.2}$ |
| subtract-reset + IF | $88.1_{\pm0.1}$ | $92.9_{\pm0.2}$ | $91.2_{\pm0.1}$ | $95.0_{\pm0.1}$ |
| zero-reset + LIF | $88.5_{\pm0.1}$ | $92.9_{\pm0.3}$ | $92.0_{\pm0.2}$ | $95.2_{\pm0.1}$ |
| subtract-reset + LIF | $88.6_{\pm0.1}$ | $92.9_{\pm0.1}$ | $92.1_{\pm0.1}$ | $\textbf{95.3}_{\pm0.2}$ |
| zero-reset + PLIF | $\textbf{89.0}_{\pm0.3}$ | $92.5_{\pm0.3}$ | $92.1_{\pm0.1}$ | $95.2_{\pm0.1}$ |
| SPIKEGCL (subtract-reset + PLIF) | $88.9_{\pm0.3}$ | $\textbf{93.0}_{\pm0.1}$ | $\textbf{92.8}_{\pm0.1}$ | $95.2_{\pm0.6}$ |

does not necessarily depend on the specific implementation details of the reset mechanisms. The performance gap between the best architecture and the worst is not significant. Even a simple IF neuron with a zero-reset mechanism can achieve satisfactory performance.

