# OpenReview forum: "A Graph is Worth 1-bit Spikes: When Graph Contrastive Learning Meets Spiking Neural Networks"
_ICLR.cc/2024/Conference — ICLR 2024 poster_

### Official Review · Reviewer_bEwP · 2023-10-30

**Soundness:** 3 good
**Presentation:** 2 fair
**Contribution:** 3 good
**Rating:** 6
**Confidence:** 3

**Summary:**

This work presents SpikeGCL which targets on optimizing GCL with SNN. They provide detail experiments to demonstrate the efficiency of the proposed method.

**Strengths:**

1.	Connect SNN and GNN is a very important topic, since GNN is closer to the neuron system and SNN is closer to the neuron dynamic.
2.	It is interesting to combine feature dimension and temporal axis.

**Weaknesses:**

1.	The detailed background does not provide in background section, and there are too many existing work descriptions in the method sections.
2.	Some design choices do not present.

**Questions:**

1.	In Sec3, does the problem formulation special for this work, or it is general for all GCL application? Please make it clearly.
2.	I think author should formulate the GCL problem in background (instead of giving a brief introduction). Also, it is better to highlight which part is optimized by the proposed methods.
3.	Why T encoders? Usually, neurons adopt the same weight among T time-steps, this design may increase the model size. Also, it is not clear how computation complex relates to time step size, since the author claim that they partition the feature dimension into T blocks. In my opinion, the computation complexity would not change when modifying T, which is not consist to Fig4.
4.	Fig3(b) ‘y1->y2’. Also, it is not clear how the entire backpropagation work. whether all y1…yn can directly receive gradient from the loss? Sec 4.4 introduces too much previous studies, it is better to clarify the gradient diagram in revision, i.e. for different blocks, where the gradient come from
5.	How SPIKEGCL can reduce parameter size? Usually, SNN can reduce the activation size but keep parameter size unchanged. Author should provide a diagram of how to compute the parameter size.

---

> ### Author Response · Authors · 2023-11-16
> **Response to Reviewer bEwP (part 1/2)**
>
> We appreciate the reviewer for detailed comments and suggestive feedback. We want to clarify some misunderstandings that caused some of your concerns.
>
>
>
> >  **W1, Q1, and Q2:** (1) Background section; (2) In Sec3, does the problem formulation special for this work, or it is general for all GCL application? Please make it clearly. (3) Author should formulate the GCL problem in background (instead of giving a brief introduction). Also, it is better to highlight which part is optimized by the proposed methods.
>
> We respectfully disagree with the point on our presentation. The current manuscript is carefully polished, as also recognized by Reviewers n3GK, jihE, and HWYD with high scores 3, 4, and 3 on the **Presentation** part. We would like to address your main concerns w.r.t. the background part below:
>
> + The problem addressed in this work is specifically formulated for binarized graph contrastive learning, with a slight modification from the general formulation of GCL. In this work, the output representations of GCL are required to be binarized.
>
> + Given the prevalence of GCL in current research, we don't think it is necessary to put a large body of context to provide a detailed formulation of the GCL problem in the paper. We believe that providing a brief introduction to GCL is sufficient to establish the context for our work on binarized graph contrastive learning, as the readers in this field are likely to be familiar with the fundamental concepts and principles of GCL.
> + In SpikeGCL, our primary focus is on optimizing the encoder part of GCL using SNNs to achieve the binarized node representation.
>
>
>
> > **Q3:** Why T encoders?
>
> We believe there are some misunderstanding on our work. We want to clarify some misunderstandings that caused some of your concerns.
>
> + **Reason to use T encoders.** The use of $T$ encoders are well-motivated in response to the increasing scale of graph data. As explained in Sec. 4.1, previous approaches for handling sequential inputs from non-temporal graphs often involve repeating the graph $T$ times with augmentation strategies.  However, this increases the computational complexity of the encoder, resulting in $T$ times the forward pass, with each step having a d-dimensional input. To address this issue, SpikeGCL utilizes $T$ encoders applied to T groups of inputs, where each group has a $\frac{d}{T}$-dimensional input. In other words, the computational complexity is reduced to approximately $\frac{1}{T}$ compared to previous methods that repeat the GNN T times. This reduction significantly decreases the computational and memory overheads associated with applying SNNs on non-temporal graphs.
>
> + **Complexity of $T$ encoders in SpikeGCL.** The use of $T$ encoders in SpikeGCL **does not** introduce additional overheads because the learnable parameters of the T encoders, **except for the first layer**, are shared across layers. Moreover, the first layer of each encoder handles only $\frac{d}{T}$ features, allowing the complexity of SpikeGCL to remain certainly low.

---

> ### Author Response · Authors · 2023-11-16
> **Response to Reviewer bEwP (part 2/2)**
>
> > Q4: Typo in Fig.3(b).
>
> Thank you for pointing it out, corrected in the revision.
>
>
>
> > Q5: It is not clear how the entire backpropagation work. whether all y1…yn can directly receive gradient from the loss? Sec 4.4 introduces too much previous studies, it is better to clarify the gradient diagram in revision, i.e. for different blocks, where the gradient come from.
>
> We apologize for any confusion caused. The gradient of each block is derived from the contrastive loss computed on the output spikes at each time step. In other words, **we calculate the contrastive loss for the outputs of each time step and perform backward propagation for each individual block.** To prevent the gradients from flowing through the blocks, we utilize stop-gradient. For a clearer illustration, we kindly refer you to Algorithm 2 in Appendix C.
>
>
>
> > Q6: How SPIKEGCL can reduce parameter size? Usually, SNN can reduce the activation size but keep parameter size unchanged. Author should provide a diagram of how to compute the parameter size.
>
> + **Parameter-shared Encoders**. As discussed in Appendix D.1, the learnable parameters of the $T$ encoders are shared across layers **except the first layer** to meet the diverse input dimensions of group of features.  The matrix dimension of learnable weight of the $T$ encoders are: layer 1 ($T \times \frac{d_1}{T}\sim d_1$), layer 2 ($d_2$), ..., layer $L$ ($d_L$), which is equivalent to a simple GNN with $d_1$, $d_2$,..., $d_L$ dimensions.
>
> + **Many-to-one Decoder**. We introduce a simple single-layer MLP as our decoder rather than other complicated architectures. The decoder is peformed on the output spikes of $T$ encoders.
> + **Small Embedding Size**: As SpikeGCL can accumulate the output spikes across $T$ time steps, it is sufficient to use a small embedding size (e.g., 16) to achieve a decent performance.
>
> In conclusion, the learnable parameters in SpikeGCL consist of a single GCN and a single MLP, both with small embedding sizes. Compared to other SNNs and GCLs that utilize complex encoders/decoders and large embedding sizes (e.g., 1024), SpikeGCL demonstrates much higher parameter efficiency.
>
>
>
> If there are any remaining concerns about the presentations of our work, please let us know. We are more than happy to provide additional information to help clarify the situation.

---

> > ### Comment · Reviewer_bEwP · 2023-11-22
> > **Response to authors**
> >
> > Thanks for the detailed response. Based on authors' reply, I have a few more questions:
> >
> > 1. Based on theorem 1, does it mean there is some redundancy in the node feature? If yes, why not directly apply compression technology on the node features, since the proposed method also has accuracy loss. Furthermore, is it necessary to identify which features should be grouped together?
> >
> > 2. Based on the explanation, I think the T encoders use identical weights, if yes please clarify this in the document.
> >
> > 3. (minor  & not necessary in revision) Based on Eq. 31, I cannot understand why the proposed method can achieve such energy saving, I think the encoding part in summation should be d/T? It would be clearer if the author could provide a table that compares the #computation and memory consumption between the proposed method and previous studies (in formula). Also, it would be better to provide the computation portion of the encoder in GNN.

---

> ### Author Response · Authors · 2023-11-23
> **Official Comment by Authors (1/2)**
>
> Thank you for the follow-up discussion and all your suggestions. Please find our responses to your latest comments below.
>
> > Based on theorem 1, does it mean there is some redundancy in the node feature?
>
> No, the core insights of Theorem 1 aim to reveal that SpikeGCL can approximate the performance of its full-precision counterpart (GNNs) with binary spikes in $T$ time steps. We draw this conclusion by demonstrating that **the SNN neuron, such as IF and LIF, in SpikeGCL serves as an unbiased estimator of the ReLU activation function of GNNs over time.**
>
> > Furthermore, is it necessary to identify which features should be grouped together?
>
> That's a good point. Honestly, we were considering employing clustering methods, such as KMeans, to group the input features. This way, each encoder can learn from a corresponding group with similar features. However, the challenge here is **how to define a new augmentation method to generate negative samples, as the clustering method is permutation-insensitive, and simple random shuffling on features does not work in this case.** Additionally, another potential issue is that grouping similar features together may pose a risk to the learning of downstream encoders, as it may lead to the oversmoothing issue. We leave them as future work.
>
> > Based on the explanation, I think the T encoders use identical weights, if yes please clarify this in the document.
>
> Not exactly. The $T$ encoders use identical weights, **except for the first layer**, to accommodate the diverse input dimensions of different groups of features. This has been clarified in Paragraph **Encoder**, Section 4.3 (page 6).

---

> ### Author Response · Authors · 2023-11-23
> **Official Comment by Authors (2/2)**
>
> > Based on Eq. 31, I cannot understand why the proposed method can achieve such energy saving, I think the encoding part in summation should be d/T? It would be clearer if the author could provide a table that compares the #computation and memory consumption between the proposed method and previous studies (in formula).
>
> Thank you for your question. Firstly, it's important to note that **Eq. 31 primarily focuses on computing the energy in SNN-based methods**. Therefore, in this response, we specifically discussion the energy efficiency of SpikeGCL in comparison to other graph SNNs.
>
> + The enhanced efficiency of SpikeGCL primarily stems from the improved encoding process.  As mentioned in the paper, prior works typically repeat the graph $T$ times to generate the sequential inputs for SNNs. This leads to high energy consumption $E_\text{encoding}$ particularly on large-scale graph datasets. In SpikeGCL, we address this issue by partitioning features into $T$ groups.
> + Following your suggestion, we provide a summary of the energy consumption of various graph SNNs in formula and conduct an empirical comparison of encoding and spiking energy consumption. The evaluation is performed with $T=30$ on the Computers dataset.The results are presented below.
> |   |     |     |     |     |     |
> |---|:---:|:---:|:---:|:---:|:---:|
> |Models     |Energy Consumption|$E_\text{encoding}$(mJ)|$E_\text{spiking}$(mJ)|$E$(mJ)||
> |SpikeNet   | $E_{MAC} \times \sum_{t=1}^{T}  NDs^t + E_{SOP} \times \sum_{t=1}^{T} S_{Sage_1}^t + S_{Sage_2}^t + S_{FC}^t$  | 0.403 | 0.031 | 0.434 ||
> |SpikingGCN | $E_{MAC} \times \sum_{t=1}^{T} NDs^t + E_{SOP} \times \sum_{t=1}^{T} S_{SGC}^t$ | 0.403 | 0.006 | 0.409 ||
> |GC-SNN     | $E_{MAC} \times \sum_{t=1}^{T} Nds^t + E_{SOP} \times \sum_{t=1}^{T} S_{GCN_1}^t + S_{GCN_2}^t + S_{SFTN_1}^t + S_{SFTN_2}^t$ | 0.403 | 0.062 | 0.465 ||
> |GA-SNN     | $E_{MAC} \times \sum_{t=1}^{T} Nds^t + E_{SOP} \times \sum_{t=1}^{T} S_{GAT_1}^t + S_{GAT_2}^t + S_{SFTN_1}^t + S_{SFTN_2}^t$ | 0.403 | 0.055 | 0.458 ||
> |SpikeGCL   | $E_{MAC} \times Nds + E_{SOP} \times \sum_{t=1}^{T} S_{GCN}^t$ | 0.013 | 0.039 | 0.052 ||
>
> + Here, $s^t$ represents the firing rate of input node features after rate-based encoding at each time step $t$, $S_{*}^t$ denotes the number of output spikes for each layers in the network at time step $t$.
> + As can be observed, SpikeGCL significantly reduces $E_\text{encoding}$ and therefore the overall energy consumption is saved.
> + As $T$ increases and the graph scales, the superiority of SpikeGCL over other graph SNNs would become more pronounced.
>
> ---
>
> Regardless, we are deeply grateful for all your previous feedback and advice, which have been invaluable to us.

---

### Official Review · Reviewer_HWYD · 2023-10-31

**Soundness:** 3 good
**Presentation:** 3 good
**Contribution:** 3 good
**Rating:** 6
**Confidence:** 2

**Summary:**

The paper presents SpikeGCL, a GCL framework built upon SNNs to learn 1-bit binarized graph representations and enable fast inference. The authors shows that Spike GCL achieves high efficiency and reduces memory consumption, and is also theoretically guaranteed with powerful capabilities to learn representations. Extensive experimental results verified that spikeGCL achieves comparable or superior performance to full-precision competitors.

**Strengths:**

The paper presents a new framework for learning on graph data, the spikeGCL. The paper is well written with clear introduction of the model and the learning algorithm to prevent the vanishing gradient problem, and presents both theoretical guarantees and extensive numerical results to demonstrate the capabilities of the model.

**Weaknesses:**

The paper focuses on the learning algorithm and performance of SpikeGCL, I think it would be interesting to further explore the properties of the 1-bit node representations themselves and compare them with other baseline models, to better understand why the learned graph representations are superior to other binary GNNs.

**Questions:**

How much does the result rely on detailed implementation of the SNN (such as reset to 0/reset by subtraction/IF or LIF)?

---

> ### Author Response · Authors · 2023-11-16
> **Response to Reviewer HWYD**
>
> We thank the reviewers for reading our paper and providing detailed review on our submission. We respond to the reviewers’ major concerns and questions below.
>
> > Q1: How much does the result rely on detailed implementation of the SNN (such as reset to 0/reset by subtraction/IF or LIF)?
>
> Thank you for your feedback. We appreciate your constructive suggestion. In the submitted version, we have included the ablation results of different spiking neurons (IF, LIF, and PLIF) in Table 7, Appendix F. Additionally, following your suggestion, we have now added the ablation results on the reset mechanism for four datasets. The results are presented below:
>
> |                               | Computers                         | Photo                             | CS                                | Physics                           |
> | ----------------------------- | --------------------------------- | --------------------------------- | --------------------------------- | --------------------------------- |
> | zero-reset + IF               | 88.0±0.2                   | 92.7±0.3                   | 91.2±0.1                   | 95.0±0.2                   |
> | subtract-reset + IF           | 88.1±0.1                   | 92.9±0.2                   | 91.2±0.1                   | 95.0±0.1                   |
> | zero-reset + LIF              | 88.5±0.1                   | 92.9±0.3                   | 92.0±0.2                   | 95.2±0.1                   |
> | subtract-reset + LIF          | 88.6±0.1                   | 92.9±0.1                   | 92.1±0.1                   | **95.3±0.2**              |
> | zero-reset + PLIF             | **89.0±0.3**               | 92.5±0.3                   | 92.1±0.1                   | 95.2±0.1                   |
> | ours (subtract-reset + PLIF) | 88.9±0.3                   | **93.0±0.1**               | **92.8±0.1**               | 95.2±0.6                   |
>
> Based on our findings, the performance does not necessarily depend on the specific implementation details of the SNN architectures. The performance gap between the best architecture and the worst is not significant. Even a simple IF neuron or a reset to 0 mechanism can achieve satisfactory performance.
>
>
>
> We would be grateful if you tell us whether the response answers your questipns of SpikeGCL, if not, what we are lacking, so we can provide better clarification. Thank you for your time.

---

> > ### Comment · Reviewer_HWYD · 2023-12-02
> >
> > The authors' response answer my question. I would like to keep my score

---

### Official Review · Reviewer_jihE · 2023-11-01

**Soundness:** 3 good
**Presentation:** 4 excellent
**Contribution:** 2 fair
**Rating:** 6
**Confidence:** 4

**Summary:**

The paper addresses the challenge of learning full-precision representations in graph neural networks,
which can be computationally and resource-intensive. The authors propose a new approach that
combines graph contrastive learning with spiking neural networks to improve efficiency and accuracy.
The proposed framework, SPIKEGCL, learns binarized 1-bit representations for graphs and provides
theoretical guarantees to demonstrate its comparable expressiveness with full-precision counterparts.

**Strengths:**

1. The motivation is clear. The paper combines graph contrastive learning with spiking neural
networks to improve efficiency and accuracy.
2. The proposed method is tested on several benchmarks.
3. The paper is well-written and provides a promising direction for graph contrastive learning with
spiking neural networks.

**Weaknesses:**

1. The author divided the original graph in time in the feature dimension and obtained T graph
structures with the same structure and reduced the node feature dimension to N/T. Compared with
copying T copies, it saves storage resources. However, the author did not explain the reason for
this approach. For example, from my personal understanding, the author's approach can be
understood as for a 1xd feature vector, there is a temporal relationship between the 0th value and
the N/T-th value, which we can’t understand.
2. The author used the SNN method to compress the original representation. One problem is that
SNN considers the accumulation in time and does not take into account the distribution
characteristics in time. What I mean is, if the characteristics of time T-1 and time T-2 are
exchanged. It seems that the value of time T will not be affected, but they will become two
completely different vectors. In this way, will there be a many-to-one situation during the
compression process?
3. I think the article lacks some quantitative analysis, such as what is the connection between the
compressed binary vector and the original vector, what is the distribution of the conventional

**Questions:**

Please see the weakness section for detailed questions.

---

> ### Author Response · Authors · 2023-11-16
> **Response to Reviewer jihE**
>
> We appreciate the reviewer for detailed comments and suggestive feedback. We want to clarify some misunderstandings that caused some of your concerns.
>
>
>
> > Q1: The feasibility of feature partitioning for SNNs.
>
> Thank you for your insightful question. While SNNs require sequential inputs, they do not necessarily require a strict temporal relationship between individual inputs like RNNs. In SNNs, the main operation is to accumulate inputs to generate spikes through an integrate-and-fire process. The occurrence of spikes carries information about the input data, where the number of spikes fired in the output of encoders reflects the importance of input feature groups.
>
> In SpikeGCL, we leverage the **spikes in all time steps** by concatenating them to obtain node representations. As a result, the precise timing or order of spikes is not always crucial as the relative timing and occurrence of spikes can still capture essential information.
>
> In the literature, there have been successful attempts, such as SpikingGCL and GC-SNN, where SNNs are employed on non-temporal graphs by augmenting them to form the sequential inputs.
>
>
>
> > Q2: If the characteristics of time T-1 and time T-2 are exchanged. It seems that the value of time T will not be affected, but they will become two completely different vectors. In this way, will there be a many-to-one situation during the compression process?
>
> There seems to be a misunderstanding regarding the mechanism of SpikeGCL. While a strict temporal relationship between inputs is not necessary, the input order does matter for SpikeGCL because the SNN neuron determines whether to fire-and-reset based on the input at the current time step. Therefore, if the characteristics of time T-1 and time T-2 are exchanged, it will affect the value at time T.
>
> Technically, SNNs are a many-to-many architecture, where each time step is associated with a fired output spike. Hence, there won't be a many-to-one situation during the compression process.
>
>
>
> > Q3: Quantitative analysis of the connection between the compressed binary vector and the original vector.
>
> Thank you for the insightful suggestion. We have included the experiments regarding the similarity between the compressed binary vector and the original input vector in **Appendix F, page 23 of the revised manuscript.** The results show that each group of features exhibits a strong correlation with the corresponding output spikes while demonstrating minimal correlation with spikes in other time steps. This suggests that the learned binary representations are disentangled from each other, thereby providing improved expressiveness in representing the input features.
>
>
>
> If any concern still remains that might prohibit a positive recommendation of this work, we would appreciate if you could let us know now.

---

### Official Review · Reviewer_n3GK · 2023-11-01

**Soundness:** 3 good
**Presentation:** 3 good
**Contribution:** 2 fair
**Rating:** 3
**Confidence:** 2

**Summary:**

This paper presents a novel graph contrastive learning (GCL) framework called SPIKEGCL, which leverages sparse and binary characteristics to learn more biologically plausible and compact representations. The proposed framework outperforms many state-of-the-art supervised and self-supervised methods across several graph benchmarks, achieving nearly 32x representation storage compression. The paper also provides experimental evaluations and theoretical guarantees to demonstrate the effectiveness and expressiveness of SPIKEGCL.

**Strengths:**

1.	This paper propose a novel GCL framework called SPIKEGCL that leverages sparse and binary characteristics to learn more biologically plausible and compact representations.
2.	This paper provides theoretical guarantees to demonstrate the expressiveness of SPIKEGCL.
3.	SpikeGCL nearly 32x representation storage compression and outperforming many state-of-the-art supervised and self-supervised methods across several graph benchmarks.
4.	Extensive experimental evaluations to demonstrate the effectiveness of the proposed framework.

**Weaknesses:**

1.	In Section 4.1, to reduce the complexity of SNNs by sampling from each node, the authors uniformly partition the node features into T groups, which is unreasonable. Features of different dimensions may represent different meanings, and operations after grouping these features may lead to inconsistencies in the feature space between different groups. On the contrary, the traditional mask method retains most features by randomly masking some features, ensuring the consistency of feature distribution. Therefore, in this section, the author can consider using random masks to reduce computational complexity while ensuring the consistency of data distribution.
2.	In table 2, the authors compare the parameter size and energy consumption between proposed method with traditional unsupervised/self-supervised mehtods. Howvere, from table 1, the performance of spikeGCL is worse than the spike-based mehtods in most cases, there’s no evidence that SpikeGCL is better than other mehtods. The authors should add the comparision between SpikeGCL with spike-based mehtods in Table 2.
3.	An intuitive question: Contrastive learning usually generates rich features from multiple perspectives to represent the target. However, spike-based methods usually lose a large amount of data, that is, a large number of learnable features are lost. Why can SpikeGCL still achieve similar results compared with traditional contrastive learning methods?
4.	Generally speaking, combining Spiking and GCL is a good idea, but the novelty is not enougt. Compared with traditional methods, SpikeGCL only groups features and then uses the traditional GCL method for learning, which does not present the special nature of contrastive learning in the scenario where spike and graph are combined.

**Questions:**

check the comments above

---

> ### Author Response · Authors · 2023-11-16
> **Response to Reviewer n3GK (part 1/2)**
>
> We thank the reviewers for reading our paper and providing detailed review on our submission. We respond to the reviewers’ concerns and questions one by one.
>
> >  Q1: (1) The authors uniformly partition the node features into T groups, which is unreasonable. (2) the author can consider using random masks to reduce computational complexity while ensuring the consistency of data distribution.
>
> We believe there are some misunderstandings on our work. To clarify the situation, here we take a simple MLP (without bias terms) as the encoder. Given an input d-dimensional vector $X$, the output $y$ is defined as: $y= X W$, where $W \in \mathbb{R}^{d \times f}$ and $f$ is the output dimension. Typically, it is equivalent to:
> $$
> \begin{align}
> y = y1 || y2 || ...||yT, \quad
> y_t= X_t W_t
> \end{align}
> $$
>
> Where $X_t$ is the $t$-th group of features with dimension $\frac{d}{T}$ and $W_t \in \mathbb{R}^{\frac{d}{T} \times f}$ is the learnable weight matrix of the $t$-th encoder. Each group of features is independently learned and this does not lead to any inconsistence in the feature space across different groups.
>
> We appreciate the reviewer‘s suggestion. While using random masks can be a valid approach to create a sequential input for SNNs, **it is not beneficial for reducing computational complexity.** In fact, it is worth noting that SpikingGCN has employed this technique to generate $T$ graphs, each with $d$-dimensional feature inputs. However, as discussed in Sec. 4.1, this method remains a significant bottleneck for SNNs when scaling to large graphs.
>
>
>
> >  Q2: From table 1, the performance of spikeGCL is worse than the spike-based mehtods in most cases, there’s no evidence that SpikeGCL is better than other mehtods. The authors should add the comparision between SpikeGCL with spike-based mehtods in Table 2.
>
> First of all, we would like to point out that the comparision between SpikeGCL with spike-based mehtods has been presented in **Table 5 of Appendix F.** In Table 5,  SpikeGCL demonstrates an average energy consumption that is approximately 50x lower, while using only 1/3 of the parameters compared to other spike-based methods.
>
> Secondly,  it is important to note that SpikeGCL is an **unsupervised (U) and binarized (B)** method, which is expected to underperform **supervised and full-precision** spike-based baselines. We understand that there is a tendency in the deep learning community to place a strong emphasis on achieving state-of-the-art performance on benchmark datasets, often measured in terms of specific performance metrics. In our work, SpikeGCL, which is an energy and memory-efficient method, demonstrates only a minor performance decrease (less than 3%) and even surpasses most supervised and full-precision spike-based baselines in 4 out of 9 datasets, as indicated in Table 1 and Table 4.
>
> While outperforming state-of-the-art methods can be an important and exciting goal for many researchers, it is also important to consider the broader context such as efficiency on the wider deep learning community. **SpikeGCL is not proposed as a new state-of-the-art SNN method. It is designed to achieve satisfied performance in low-power and resource-constrained settings, addressing the challenges posed by the increasing scale of real-world graph data.**

---

> ### Author Response · Authors · 2023-11-16
> **Response to Reviewer n3GK (part 2/2)**
>
> >  Q3: Spike-based methods usually lose a large amount of data, that is, a large number of learnable features are lost. Why can SpikeGCL still achieve similar results compared with traditional contrastive learning methods? Why can SpikeGCL still achieve similar results compared with traditional contrastive learning methods?
>
> We acknowledge that spike-based methods (SNNs) typically involve a loss of data compared to traditional rate-based methods employed in artificial neural networks (ANNs). This discrepancy arises from the fact that SNNs represent information using discrete spikes or events rather than continuous firing rates. However, it is worth noting that there have been studies[1,2] demonstrating that SNNs can approximate a ReLU network when given reasonably large time steps $T$.
>
> In our work, we also provide theoretical guarantees regarding the expressiveness of SpikeGCL. Specifically, we show that SpikeGCL has the capability to (almost) approximate a comparable GNN when utilizing a large time step $T$. As a result, SpikeGCL has the potential to achieve comparable results to traditional GCL methods. Please kindly refer to Sec. 5 and Appendix A for further detailes.
>
> [1] Rueckauer B, Lungu I-A, Hu Y, Pfeiffer M and Liu S-C (2017) Conversion of Continuous-Valued Deep Networks to Efficient Event-Driven Networks for Image Classification. Front. Neurosci. 11:682.
>
> [2] Rueckauer, B., Lungu, I. A., Hu, Y., & Pfeiffer, M. (2016). Theory and tools for the conversion of analog to spiking convolutional neural networks. arXiv preprint arXiv:1612.04052.
>
>
>
> > Q4: Novelty of SpikeGCL.
>
> We respectfully disagree with the reviewer that the novelty of our paper is limited. Here we would like to reiterate the three research contributions of our work:
>
> - **[Novelty 1]** The first-ever binary GCL method leveraging the sparse and energy-efficient characteristics of SNNs.
> - **[Novelty 2]** The theoretical analysis about why SpikeGCL can approximiate the performance of full-precision GCLs.
> - **[Novelty 3]** The block-wise training scheme on SNNs prevent them from suffering from the notorious problem of vanishing gradients with large time steps $T$.
>
> Particularly, **Novelty 2 and 3 are built upon the special nature of SNNs and are proposed to solve the unique challenges in the context of GCL.** Our work digs deeper into the SNNs in GCL and offers more insights that would potentially benefit the whole community, which could motive more interesting research in this area. We believe this is an interesting and insightful work and hope you could reevaluate the contributions of our work. If there are any remaining concerns about the novelty of our work, please let us know. We are more than happy to provide additional information to help clarify the situation.
>
>
>
> Thank you again for taking the time to review our paper. We hope our responses could clarify your concerns, and hope you will consider increasing your score. If we have left any notable points of concern unaddressed, please do share and we will attend to these points.

---

### Author Response · Authors · 2023-11-20
**A friendly reminder on the approaching discussion deadline**

Dear Reviewers,

We sincerely thank the reviewer for the constructive comments.

We have carefully considered your comments and incorporated some  discussion into the manuscripts. The updated contents are highlighted in blue. Here is a summary of the key revisions:

1. Connections between original input features and output spikes (Appendix F, page 22).
2. Ablation results on different reset mechanisms and spiking neurons (Appendix F, page 23).
3. Typo in Figure 3(b), page 7.

**The discussion period is coming to an end, and we hope that we have addressed all of your comments. Could you take a look at our responses and updated paper, and let us know if you have any follow-up questions?** We will be happy to answer them.

Kind regards,

the Authors

---

### Author Response · Authors · 2023-11-22
**Dear ACs, Could you take a chance to remind the reviwers back to discussion? Thank you so much!**

Dear ACs and Reviewers,

Thanks a lot for arranging the reviewing process so far!

We genuinely thank all the reviewers for their very helpful comments. We have been pretty active in addressing their concerns and keeping adding new results in the past few days. We would like to ask for more discussion, yet it seems the reviewers are a bit inactive in responding to our further feedback.

We deeply understand it is never an easy job to review a paper. But to ensure a fair evaluation of a paper, it might be better if the reviewers can further participate in the discussion. We are wondering if ACs could help with this. Thank you very much!

Kind regards,

the Authors

---

> ### Comment · Area_Chair_1Y5M · 2023-11-22
>
> Dear Authors,
>
> Thank you for your comment. Please rest assured that the reviewers have been nudged repeatedly to engage in the discussion, and that both reviews and author responses will be taken into account for the outcome of the review process.
>
> Kind regards,
> --Your AC

---

### Author Response · Authors · 2023-11-23
**General Response to ACs and Reviewers**

Dear ACs and Reviewers,

We sincerely thank for your valuable comments and your time in reading our paper. **We confidently believe that this work is worthy of being accepted by ICLR, because it advances the neuromorphic computing (SNN) on graph data in a tangible way, without bells and whistles.**

In this work, we have pushed the field further, **our contributions** include:

- **[C1] New architecture.** The **first-ever** binary GCL leveraging the sparse and energy-efficient characteristics of SNNs.
- **[C2] Theoritical insight.** The theoretical analysis providing insights to bridge the gap between SNNs and GNNs.
- **[C3] Training scheme.** The block-wise training scheme for SNNs to prevent them from the notorious problem of vanishing gradients with large time steps.
- **[C4] Performance.** SpikeGCL is able to achieve performance on par with advanced baselines using full-precision or 1-bit representations while demonstrating significant efficiency advantages in terms of **parameters, speed, memory usage, and energy consumption.**

We believe this is an interesting and insightful work and really hope that the ACs and the Reviewers can give more support to this new yet promising research direction.

Best regards,

the Authors

---

### Meta-Review · Area_Chair_1Y5M · 2023-12-05

**Metareview:**

This paper introduces SpikeGCL, a method for energy-efficient graph representation learning that combines graph contrastive learning (GCL) with spiking neural networks to improve efficiency and accuracy.

The reviewers primarily agree that the paper is generally well written and also highlight that it covers an important topic, that it provides a promising new direction and that the method is well-tested.

One primary concern raised by reviewer n3GK, who recommends rejection, is that the mere combination of GCL with ideas from spiking neural networks is insufficient in terms of novelty. This is a valid concern, especially since several papers on graph representation learning and spiking neural networks exist which use a very similar formulation (which the authors cite and compare to). The technical novelty can indeed be viewed as a small step beyond prior work, but I agree with the majority of the reviewer who view this work as an interesting (yet small) step forward. The paper further extensively evaluates the benefits of their contribution. In their rebuttal, I believe that the authors have sufficiently addressed this and other concerns.

Overall, I believe this paper is very much borderline for ICLR, but I think it can nonetheless be accepted.

**Justification For Why Not Higher Score:**

The paper covers a relatively niche topic, which ultimately requires special types of hardware accelerators to live up to its potential.

**Justification For Why Not Lower Score:**

Solid contribution of relevance to the ICLR community; well-written paper; claims are satisfactorily evaluated.

---

### Decision · Program_Chairs · 2024-01-16

Accept (poster)